# Learning speed and detection sensitivity controlled by distinct cortico-fugal neurons in visual cortex

**Sarah Ruediger[1,2,3]\*, Massimo Scanziani[1,2,3]\***

[1]Center for Neural Circuits and Behavior, Neurobiology Section and Department of Neuroscience, University of California, San Diego, La Jolla, United States; [2]Department of Physiology, University of California, San Francisco, San Francisco, United States; [3]Howard Hughes Medical Institute, University of California, San Francisco, San Francisco, United States

**Abstract** Vertebrates can change their behavior upon detection of visual stimuli according to the outcome their actions produce. Such goal-directed behavior involves evolutionary conserved brain structures like the striatum and optic tectum, which receive ascending visual input from the periphery. In mammals, however, these structures also receive descending visual input from visual cortex (VC), via neurons that give rise to cortico-fugal projections. The function of cortico-fugal neurons in visually guided, goal-directed behavior remains unclear. Here, we address the impact of two populations of cortico-fugal neurons in mouse VC in the learning and performance of a visual detection task. We show that the ablation of striatal projecting neurons reduces learning speed, whereas the ablation of superior colliculus projecting neurons does not impact learning but reduces detection sensitivity. This functional dissociation between distinct cortico-fugal neurons in controlling learning speed and detection sensitivity suggests an adaptive contribution of cortico-fugal pathways even in simple goal-directed behavior.

**\*For correspondence:**
sarruedi@gmail.com (SR);
massimo@ucsf.edu (MS)

**Competing interests:** The authors declare that no competing interests exist.

## Introduction

Visual stimuli guide the behavior of many animals. While the detection of ethologically relevant visual stimuli can elicit innate behavior, often visual stimuli become relevant through learning, leading to goal-directed behavior upon stimulus detection (*Morris et al., 2018*; *Schultz, 2006*). For example, many vertebrates can learn to alter their behavior in response to the detection of arbitrary visual stimuli in order to obtain a reward or avoid punishment (*Llinás, 1976*; *Prusky and Douglas, 2004*; *Skinner, 1963*; *Valente, 2012*). A major challenge to our understanding of the neuronal basis of this elemental form of sensory-based and goal-directed behavior is that any visual stimulus evokes neuronal activity across multiple brain structures (*Macé et al., 2018*; *Seabrook et al., 2017*) and, within each structure, across diverse types of neurons (*Harris and Mrsic-Flogel, 2013*; *Harris and Shepherd, 2015*; *Reinhard et al., 2019*).

The striatum and the optic tectum are two evolutionary conserved subcortical structures involved in the learning and performance of simple goal-directed behavior in many vertebrates. The striatum is fundamental for reinforcement learning (*Cox and Witten, 2019*) and action initiation (*Klaus et al., 2019*) and the optic tectum (called the superior colliculus in mammals) plays an important role in the detection of salient visual stimuli that trigger innate behaviors (*Feinberg and Mallatt, 2019*; *Grillner and El Manira, 2020*). At least in mammals, both structures receive two main sources of visual information: one ascending from the periphery (*Klaus et al., 2019*; *Krauzlis et al., 2013*) and the other, descending from visual cortex (VC) via its prominent cortico-fugal pathways (*Feinberg and Mallatt, 2019*; *Smeets et al., 2000*; *Suryanarayana et al., 2020*). Whether the

neurons in VC that give rise to these cortico-fugal pathways play a role in simple goal-directed behavior remains poorly understood. In fact, mammals are able to learn and perform simple sensory detection tasks even in the absence of sensory neocortex (*Ceballo et al., 2019*; *Dalmay et al., 2019*; *Hong et al., 2018*; *Pöppel et al., 1973*; *Popper and Fay, 1992*; *Weiskrantz et al., 1974*). Thus, it remains unknown whether cortico-fugal pathways, the main output pathways by which VC can influence the rest of the brain and thus, ultimately, behavior, do actually contribute to simple goal-directed behavior.

Here, we determined the impact of two major populations of cortico-fugal neurons in VC in the speed of learning and in the performance sensitivity of a visual detection task in mice. Using an intersectional viral approach to selectively eliminate specific populations of cortico-fugal neurons in VC, we show that the ablation of neurons projecting to the striatum reduces learning speed during task acquisition, whereas the ablation of neurons projecting to the superior colliculus impairs detection sensitivity during task execution. Furthermore, we show that, with training, both populations of cortico-fugal neurons eventually become dispensable for the task. These data demonstrate the functional dissociation between two distinct populations of cortico-fugal neurons in VC during specific training stages of a visual detection task and highlight that specific cortico-fugal pathways adaptively contribute even to simple goal-directed behavior.

## Results

Distinct populations of neurons in VC send visual information to the dorso-medial striatum (dmSt) and the superior colliculus (SC; the mammalian optic tectum) via two prominent cortico-fugal pathways, the cortico-striatal (CSt; *Faull et al., 1986*; *Kemp and Powell, 1970*; *Khibnik et al., 2014*; *Saint-Cyr et al., 1990*) and the cortico-tectal (CT; *Wang and Burkhalter, 2013*; *Zingg et al., 2017*), respectively (*Hattox and Nelson, 2007*; *Jones, 1984*; *Lur et al., 2016*; *Norita et al., 1991*; *Rhoades et al., 1985*; *Serizawa et al., 1994*; *Swadlow, 1983*; *Tang and Higley, 2019*). To determine the role of these two populations of cortico-fugal neurons in a simple visual detection task, we trained water restricted, head fixed mice to report the appearance of a visual stimulus with a lick on a waterspout. The visual stimulus consisted of a 30° drifting grating patch presented at full contrast (unless specified otherwise) on a computer monitor placed to the left visual field of the animal (*Figure 1A*). To be rewarded with water, the animal had to report the presence of the stimulus such that the first lick occurred during the stimulus presentation. To determine the performance of the animal, we used both the first lick latency and the probability of a lick during either stimulus presentation or blank periods. Hits were stimulus trials on which the animal licked; correct rejections were blank trials (no stimulus) in which the animal refrained from licking throughout the response period. Misses and false alarms were omission of licks on stimulus trials, and licks on blank trials, respectively (*Figure 1A*; for trial structure see *Figure 1—figure supplement 1*). To determine the visual specificity of the licking behavior, that is the extent to which licking is guided by the visual stimulus, we compared the first lick latency distributions between stimulus and blank trials using Receiver Operating Characteristics (ROC) analysis (*Fawcett, 2006*; *Macmillan and Creelman, 2005*). The area under the ROC curve (aROC) reports the probability of an ideal observer to correctly classify trial type (i.e. stimulus versus blank) based on first lick latency. An aROC of 1 corresponds to a complete separability of the two trials types based on the distribution of first lick latencies, and thus a perfect classification. In contrast, an aROC of 0.5 represents a complete overlap of the temporal distributions of first lick latencies and thus, chance performance. With the aROC of first lick latencies, it is thus possible to detect the emergence of visually guided behavior even while hits and false alarms occur at the same rate, as long as the first lick latency in response to a stimulus trial differs from that to a blank trial.

Learning was characterized by a gradual increase in the aROC (linear fit from day 1 to day 14 aROC/day: $0.028 \pm 0.003$; mean $\pm$ SEM; N = 8 mice; population average slope aROC/day: $0.030 \pm 0.023$; mean $\pm$RMSE, $R^2 = 0.9584$, p<0.001; F-test), by a progressive reduction in first lick latency and a reduction in the trial-to-trial variability of the first lick latency on stimulus trials (*Figure 1C*). Learning was considered complete when it reached an aROC of at least 0.8 for 4 consecutive days. On average learning was completed by the 14th day of training (aROC $0.86 \pm 0.04$; first lick latency $0.52 \pm 0.13$ s; variability:$0.12 \pm 0.05$ s; mean $\pm$ SEM; N = 8 mice). The temporal distributions of first lick latencies started to differ between stimulus and blank trials early during training

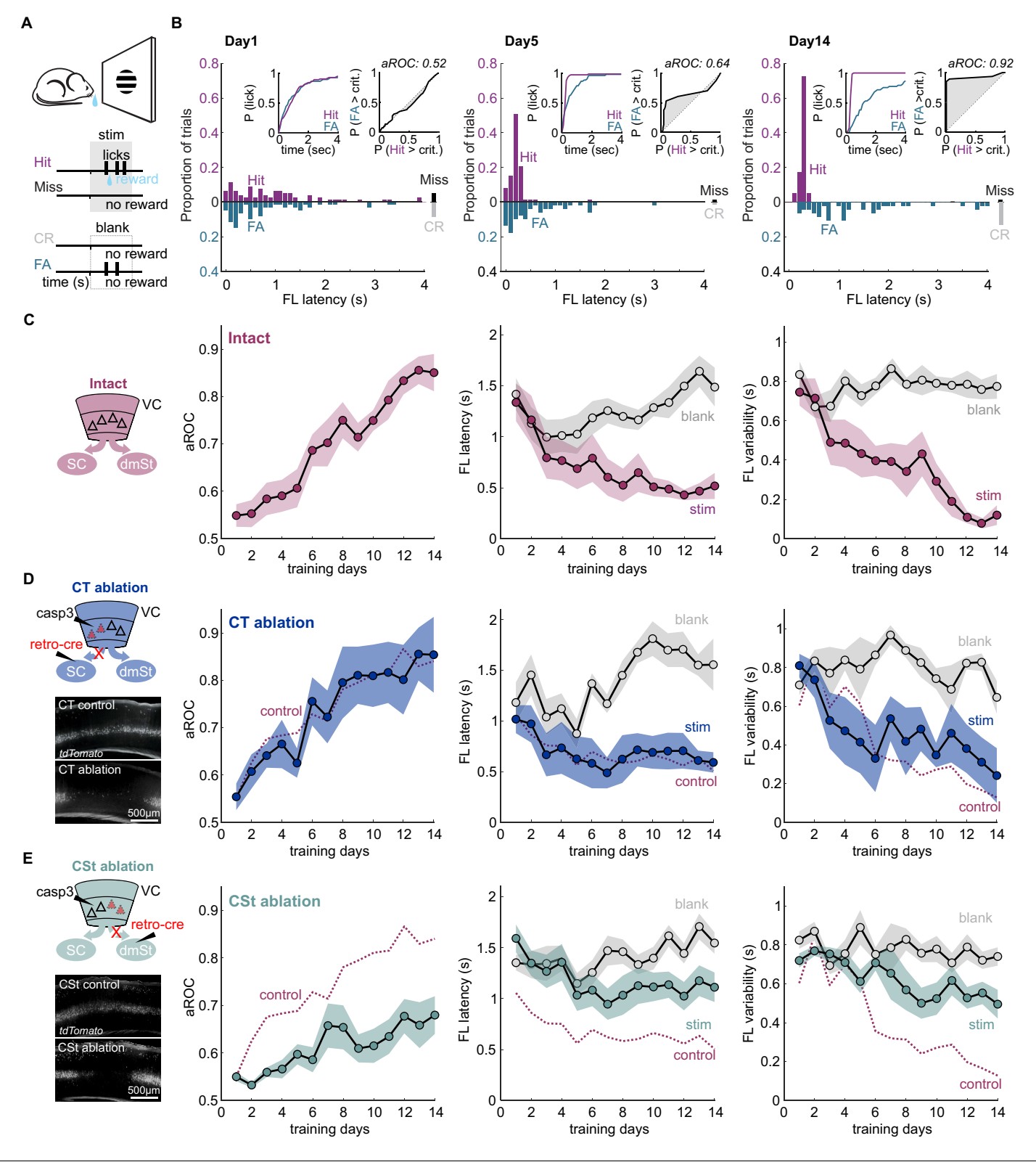

**Figure 1.** Ablation of cortico-striatal neurons impairs learning of a simple visual detection task. (**A**) Schematic of behavioral setup: mice have to lick in response to a visual stimulus to obtain reward (Hit) and omit licking on blank trials (Correct Rejection: CR). False Alarms (FA) and misses (Miss) are not punished. (**B**) Data from example mouse on days 1, 5, and 14 of training. First lick (FL) latency in response to stimulus (Hit) and a corresponding blank period (FA) during the 4 s response period. The proportion of misses (Miss) and CRs is shown on the right of each plot. Inset: Left: Cumulative

*Figure 1 continued on next page*

*Figure 1 continued*

probability of licking for stimulus trials (Hit) and corresponding blank periods (FA). Right: Area under the Receiver Operating Characteristic Curve (aROC) for first lick latency distributions on stimulus trials (Hit) and during corresponding blank periods (FA) relative to criterion (crit.). Note that by day 5 the distribution of FL latencies on stimulus trials (Hit) is already shifted toward shorter intervals as compared to that on blank trials (FA) indicating faster stimulus-guided responses on Hit trials., The probability of a FA increases gradually as time elapses. (C) Left: schematic of visual cortex (VC) with two intact cortico-fugal pathways to the superior colliculus (SC) and dorsomedial striatum (dmSt). Right: Population average learning curves over 14 days of training (N = 8 mice). aROC (left), FL latency for stimulus and blank trials (middle), FL variability for stimulus and blank trials (right; see Materials and methods). Data plotted as mean ± SEM. (D) Left: Schematic of viral approach to ablate cortico-tectal (CT) neurons. Experiments were performed in Ai14 animals to conditionally express td-Tomato in neurons infected with retroAAV-Cre (retro-cre). Red triangles with dashed borders are retrogradely transfected neurons conditionally expressing td-Tomato and Caspase 3. Below: Fluorescence microscopy image of a control animal (not injected with AAV-Casp3 in VC; top) and of an animal injected with AAV Casp3 in VC (bottom). Note the strong reduction in fluorescence in layer five at the injection site. Right: Population average learning curves as in (C) but for mice in which the CT pathway was ablated before training onset (N = 5 mice). The dotted line is data from control mice receiving an injection of retroAAV-Cre but not AAV-Casp3 (N = 7; *Figure 1—figure supplement 6*). (E) As in (D) but for mice with ablation of cortico-striatal (CSt) neurons (N = 5 mice). The dotted line is data from control mice, as in (D). Data plotted as mean ± SEM. Note the learning impairment in CSt-ablated animals.

The online version of this article includes the following source data and figure supplement(s) for figure 1:

**Source data 1.** Behavioral performance measurements as a function of training days for intact animals, CT-ablated, and CSt-ablated animals.
**Figure supplement 1.** Trial structure of the detection task.
**Figure supplement 2.** Comparison of aROC analysis with probability of licking and d-prime.
**Figure supplement 2—source data 1.** Comparison of performance metrics including aROC, probability of licking and d-prime.
**Figure supplement 3.** CT and CSt pathways originate from two separate neuronal populations.
**Figure supplement 3—source data 1.** Histological measurements.
**Figure supplement 4.** Comparing CTB with retroAAV for retrograde labeling.
**Figure supplement 5.** Histology of cortico-fugal pathway ablations.
**Figure supplement 5—source data 1.** Histological data for CSt ablation and CT ablation.
**Figure supplement 6.** RetroAAV-Cre does not impair learning.
**Figure supplement 6—source data 1.** Behavioral performance measurements for animals injected with retroAAV-Cre only.

(*Figure 1B* and *Figure 1—figure supplement 1*) revealing the beginning of the formation of the association between stimulus and reward by the animal (day 4 FL latency: stim. 0.77 ± 0.20 s vs. blank: 1.01 ± 0.16 s; mean ± SEM; N = 8 mice; p<0.01; Wilcoxon signed-rank test; *Figure 1C* and *Figure 1—figure supplement 2*). In contrast to ROC analysis, lick probability and thus d-prime strongly depended on the duration of the considered response window (*Figure 1B* and *Figure 1—figure supplement 2A*). For example, by day 4 of training, while aROC was well above chance (aROC = 0.60 ± 0.03; mean ± SEM; N = 8 mice; p<0.001; Mann-Whitney U test), there were as many hits as false alarms over the full 4 s response window (Day 4: Hit Rate: 89.9% ± 3.8 False Alarm Rate: 81.6% ± 4.9; mean ± SEM; N = 8 mice; n.s.; Wilcoxon signed-rank test; d-prime: 0.6 ± 0.13, *Figure 1—figure supplement 2A–B*). That is, the animals licked earlier in the response window following a stimulus as compared to a blank (Day 4: mean FL latency Stim vs Blank), but the lick probability within the whole response window was similar for the two conditions, mainly because of the high spontaneous lick rate of animals early in learning. Accordingly, reducing the time window within which to analyze licks probabilities increased d-prime, up to the 'optimal' response window, where d-prime is maximal (d-prime based on optimal response window per animal: 1.49 ± 0.48 s; mean ± SEM; N = 8 mice; *Figure 1—figure supplement 2B*). The optimal response window, however, varied widely across training days and animals (shortest optimal response window 0.38 ± 0.07 s vs longest optimal response window: 2.90 ± 0.39 s; mean ± SEM, N = 8 mice; p<0.001; Wilcoxon signed-rank test; average optimal response window: 1.19 ± 0.19 s; *Figure 1—figure supplement 2B*). In contrast, by defining proficiency based on the temporal distribution of first lick latencies using the aROC rather than on the probability to lick, mice were able to reach expertness before maximizing inhibitory control throughout the 4 s response window. Thus, aROC analysis over an extended response window captures the temporal characteristics of the behavioral response and offers a reliable performance metric to determine the specificity of stimulus guided responses.

To address the contribution of VC neurons projecting to the SC or to the dmSt to the learning of this simple detection task, we selectively ablated CT or CSt neurons, respectively. For this, we used an intersectional viral strategy that takes advantage of a designer AAV-Cre virus (*Madisen et al., 2015*; *Tervo et al., 2016*) to conditionally express taCaspase3 (taCasp3; *Yang et al., 2013*) in select

populations of cortico-fugal neurons. Using the retrograde tracer Cholera toxin B (CTB; *Luppi et al., 1990*; *Wan et al., 1982*), we verified that CT and CSt neurons indeed represent distinct populations in VC (overlap <3.4%; 107 of 3187 CTB-labeled cells; 4.62 ± 1.13% of the CSt and 8.82 ± 4.88% of the CT population; *Figure 1—figure supplement 3*), consistent with previous reports (*Brown and Hestrin, 2009*; *Lur et al., 2016*; *Norita et al., 1991*; *Serizawa et al., 1994*). The vast majority of CT and CSt neurons were located in layer 5 and, while the two populations of neurons largely intermingled, CT neurons tended to be more tightly distributed toward the lower part of the layer (*Figure 1—figure supplement 3*).

To determine whether CT neurons are necessary to learn the detection task, we selectively ablated them by injecting, 3 weeks before training onset, Ai14 reporter mice with retroAAV-Cre (*Tervo et al., 2016*) in the right SC to conditionally express taCasp3 (*Yang et al., 2013*) in the right VC, contralateral to the visual stimulus. The efficiency of retroAAV-Cre in retrogradely labeling cortico-fugal neurons was similar to CTB (retroAAV-Cre: 30 ± 14 cells/100 µm$^3$; retrograde tracer CTB: 24 ± 9 cells/100 µm$^3$ in layer 5; mean ± SD; N = 3 mice). We also directly compared the efficiency of the retroAAV-Cre by co-injecting it with CTB in the Rosa26 LSL H2B reporter mouse. The retroAAV-Cre was highly efficient as the majority of CTB-labeled cells co-expressed nuclear mCherry (85.5%, 864 cells of 1010, N = 2 mice; *Figure 1—figure supplement 4*). The viral ablation of Cre recombinase expressing CT neurons in VC was highly efficient with over 90% of ablated neurons 3 weeks after the injection, as verified histologically (retroAAV-Cre only: 26 ± 8 cells/100 µm$^3$ in layer 5 vs. retroAAV-Cre and AAV-taCasp3: 1 ± 1 cells/100 µm$^3$ in layer 5; mean ± SD; N = 3 mice; p<0.001 Mann-Whitney U test; *Figure 1—figure supplement 5A*). The expression of taCasp3 reduced the population of NeuN-labeled neurons in layer 5 by 20.06 ± 3.95% (NeuN-labeled neurons control: 27.55 ± 1.25 cells/100 µm$^2$ vs. casp3: 21.88 ± 0.32 cells/100 µm$^2$; mean ± SEM; N = 3 mice, p<0.01 Mann-Whitney U test; cells per animal: control: 1203.2 ± 13.52; casp3: 853.7 ± 3.34; mean ± SEM). This is in line with the number of CT neurons in this layer (calculated using retrogradely labeled CTB neurons 4.04 ± 0.55 cells/100 µm$^2$; N = 3 mice) and consistent with taCasp3 toxicity being limited to Cre-expressing neurons (*Gray et al., 2010*).

Ablation of CT neurons had little impact on task learning as compared to controls in which only the retroAAV-Cre had been injected (aROC/day CT-ablation: 0.03 ± 0.006 vs. retroAAV-Cre only: 0.03 ± 0.004; mean ± SEM; N = 5 mice, n.s. Mann-Whitney U test; population average CT-ablation slope aROC/day: 0.029 ± 0.0014 vs. retroAAV-Cre only slope aROC/day: 0.026 ± 0.001; mean ±RMSE; n.s.; Fisher z-test; *Figure 1D* and *Figure 2D*). Accordingly, by the 14th day of training, CT-ablated mice reached an aROC similar to controls animals (CT-ablation aROC:0.85 ± 0.08 vs. retroAAV-Cre only aROC 0.84 ± 0.04; mean ± SEM; n.s. Mann-Whitney U test; *Figure 1D* and *Figure 1—figure supplement 6*). Consistent with the lack of impact on visual specificity of the licking behavior, also the first lick latency and variability decreased with training in CT-ablated animals as in controls (*Figure 1D* and *Figure 1—figure supplement 6*). Thus, the ablation cortico-fugal neurons that project to the SC has no effect on the acquisition of the task.

To determine whether neurons that project to the dmSt affect the ability to learn the detection task, we selectively ablated them using the same intersectional approach used for ablating CT neurons, this time however, by injecting the retroAAV-Cre in the dmSt. The viral ablation of Cre recombinase expressing CSt neurons in VC was also highly efficient with over 90% of ablated cells 3 weeks after the injection, as verified histologically (retroAAV-Cre: 20 ± 6 cells/100 µm$^3$ CSt-ablation: 1 ± 1 cells/100 µm$^3$ in layer 5; mean ± SD; N = 3 mice; p<0.001 Mann-Whitney U test; *Figure 1—figure supplement 5*). In striking contrast to control and CT-ablated mice, in CSt-ablated mice the visual specificity of the licking behavior increased much slower (aROC/day CSt-ablation: 0.014 ± 0.003 vs. retroAAV-Cre only: 0.03 ± 0.004; mean ± SEM; p<0.05; Mann-Whitney U test; population average CSt-ablation slope aROC/day: 0.014 ± 0.0018 vs. retroAAV-Cre only slope aROC/day: 0.026 ± 0.001; mean ±RMSE; Fisher z-test; p<0.001; CSt-ablation slope aROC/day: 0.014 ± 0.0018 vs. CT-ablation slope aROC/day: 0.029 ± 0.0014; mean ±RMSE; Fisher z-test; p<0.001; *Figure 1E* and *Figure 2D*). As a consequence, by day 14 of training, CSt-ablated mice had an aROC of only 0.68 ± 0.04 compared with 0.84 ± 0.04 in control animals in which only the retroAAV-Cre had been injected (p<0.001 Mann-Whitney U test; *Figure 1E* and *Figure 1—figure supplement 6*). Furthermore, both the latency and trial-to-trial variability of the first licks on stimulus trials decreased slower in CSt-lesioned as compared to retroAAV-Cre control animals (*Figure 1E* and *Figure 1—figure supplement 6B*). The spontaneous lick frequency was similar between control and CSt-ablated mice across

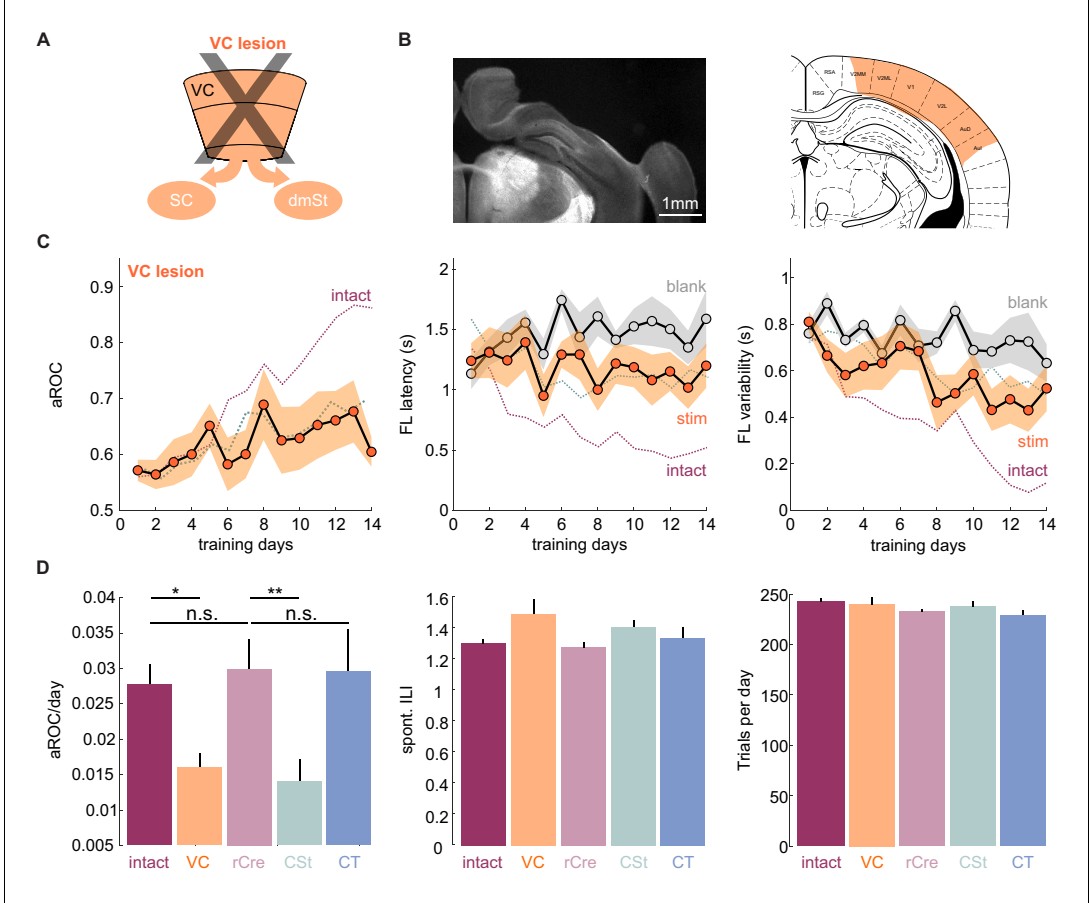

**Figure 2.** VC lesions recapitulate learning impairment induced by the ablation of cortico-striatal neurons. (**A**) Schematic of surgical ablation of visual cortex (VC). (**B**) Left: Coronal sections from an example mouse illustrating the surgical ablation of the right VC. Right: Corresponding coronal plane from the Paxinos mouse brain atlas with the spread of the lesion in orange shading. (**C**) Population average learning curves over 14 days of training for mice in which VC was surgically ablated before the onset of training (N = 8 mice). The purple dotted line is data from intact animals (from *Figure 1C*) for comparison. Green dotted line is data from CSt-ablated animals (from *Figure 1E*) for comparison. Data plotted as mean ± SEM. Note that the VC lesion recapitulates the learning impairment of CSt-ablated mice. (**D**) Left: Population average slope aROC/day across experimental groups. Intact (gray), VC lesion (VC: orange), retroAAV-Cre only (rCre: purple), CSt-lesion (CSt:aqua), CT-lesion (CT: blue). Middle: Population average of spontaneous licking (ILI: inter lick time interval) behavior during the gray screen period of the inter trial interval (ITI). Right: Population average number of trials per training session. Data plotted as mean ± SEM.

The online version of this article includes the following source data and figure supplement(s) for figure 2:

**Source data 1.** Behavioral performance measurements as a function of training days during task acquisition for VC-lesioned animals and comparison of behavioral metrics across experimental groups.

**Figure supplement 1.** Histology of cortical lesions.

**Figure supplement 1—source data 1.** Histological data to characterize the VC lesion.

the 2 weeks of training (inter-lick-interval: intact: 1.29 ± 0.03 s; retroAAV-Cre control: 1.27 ± 0.03 s vs. 1.40 ± 0.04 s mean ± SEM; n.s.; Mann-Whitney U test, *Figure 2D*). In addition, the impairments in task learning was not due to a decrease in the number of trials the animals performed during the training period neither to a reduced spontaneous lick frequency because these behavioral variables were similar in CSt-ablated and control mice (number of trials per training session: intact: 242.5 ± 3.2; retroAAV-Cre only: 232.8 ± 2.6 vs. CSt-ablation: 237.6 ± 5.3; mean ± SEM, n.s. Mann-Whitney U test; *Figure 2D*). Thus, ablation of cortico-fugal neurons that project to the dmSt led to slower learning such that by 2 weeks of training, learning was still incomplete.

The above result suggests that CSt neurons specifically contribute to the learning speed of the animal. Alternatively, CSt neurons may be required for learning itself and slow learning in CSt-ablated animals could simply result from an incomplete ablation of neurons projecting to the dmSt.

To completely eliminate the CSt neurons, we surgically removed the entire VC, that is, primary VC and the surrounding higher visual areas contralateral to the stimulus 10 days before starting behavioral training (in three out of eight animals both ipsi and contralateral VC were ablated; *Figure 2* and *Figure 2—figure supplement 1*). Both the spontaneous lick frequency and the number of performed trials per session were similar between control and VC-lesioned mice (inter lick time interval day 1: control: 0.95 ± 0.14 s; vs. VC lesion: 0.99 ± 1.47 s; mean ± SEM; number of trials per training session: control 244.7 ± 4.0 vs. VC lesion 239.3 ± 9.9 mean ± SEM; n.s. Mann-Whitney U test; *Figure 2D*). Like in CSt-ablated animals, VC lesions led to slower learning (population average VC lesion slope aROC/day: 0.014 ± 0.0014 vs. CSt ablation slope aROC/day: 0.014 ± 0.0018; mean ±RMSE; n.s.; Fisher z-test) as compared to intact animals (VC lesion slope aROC/day: 0.014 ± 0.0014 vs. control slope aROC/day: 0.025 ± 0.0014; mean ±RMSE; p<0.001 Fisher z-test) such that by 2 weeks of training none of the VC-lesioned mice reached criterion (VC lesion: aROC of 0.60 ± 0.03 vs. intact: 0.86 ± 0.04; mean ± SEM; p=0.0040; Mann-Whitney U test; *Figure 2*). Thus, VC lesions recapitulate the effect of CSt neuron ablation. These results indicate that ablation of CSt neurons impairs learning speed.

If ablation of CSt neurons selectively impairs learning speed without limiting the animal's ability to learn, it should be possible, with additional training, for CSt-ablated animals to eventually reach the same visual specificity as control animals. We thus determined the impact of additional training on animals in which, before the onset of training, we either ablated CSt neurons or lesioned VC and compared their performance with control animals. As described above, by the 14th day of training, intact animals had reached the learning criterion while CSt-ablated and VC-ablated animals were still below criterion. Importantly, by the 14th day of training, intact animals had reached plateau because further training did not improve their performance (slope aROC/day $2.46 \times 10^{-4} \pm 0.021$; mean ±RMSE; $R^2 = 9.51 \times 10^{-4}$, n.s.; F-test; population average linear fit from day 14 to day 21; *Figure 3A*). In striking contrast to the plateau performance of intact animals, both CSt-ablated and VC-lesioned animals continued to improve in the visual specificity of their licking behavior with additional training (CSt-ablation lesion slope aROC/day 0.033 ± 0.054, mean ±RMSE; $R^2 = 0.83$, F-test; p<0.001; VC lesion slope aROC/day 0.022 ± 0.063, mean ±RMSE; $R^2 = 0.78$, F-test; p<0.001; *Figure 3B*). By the end of the third week, CSt-ablated and VC-ablated animals reached an aROC of 0.87 ± 0.02 and 0.76 ± 0.05, hence similar to the plateau levels of control animals (intact aROC: 0.84 ± 0.03 vs. VC-lesion: 0.76 ± 0.05; n.s.; retroAAV-Cre only: 0.81 ± 0.05 vs. CSt-ablation: 0.87 ± 0.02 n.s., Mann-Whitney U test; *Figure 3B*).

If ablation of CSt neurons before task acquisition specifically impairs learning speed, ablation of CSt neurons in animals that master the task, should not affect performance. To address this question, we completely eliminated CSt neurons in VC, as above, by surgically removing VC in mice who had learned the detection task and tested them after a 10 days training gap following the surgery. Strikingly, mice that had learned the task before VC lesion maintained a high visual specificity of the licking behavior even after VC lesions (aROC pre-lesion: 0.92 ± 0.02 vs. aROC post-lesion: 0.91 ± 0.08, mean ± SEM; N = 5 mice, n.s. Wilcoxon signed-rank test; *Figure 4A*). VC-lesioned animals also showed no change in first lick latency (pre-lesion: 0.58 ± 0.10 s vs. post-Lesion: 0.45 ± 0.04; mean ± SEM; n.s. Wilcoxon signed-rank test) and in its variability (pre-lesion:0.54 ± 0.22 s vs. post-lesion:0.49 ± 0.24 s; mean ± SEM; n.s. Wilcoxon signed-rank test; *Figure 4—figure supplement 1*). Thus, once learning has occurred, CSt neurons are no longer required for the animal's ability to execute the task.

What is the function of the CT neurons in the detection task? While CT neurons are not involved in learning, they may play a role in stimulus detection, given the role of the SC, the main target of CT neurons, in detecting visual stimuli that elicit innate behavior (*Evans et al., 2018*; *Liang et al., 2015*; *Shang et al., 2015*; *Shang et al., 2018*). To assess the animal's sensitivity in detecting visual stimuli, we tested their performance by presenting full field visual stimuli at various contrasts. To be able to detect potentially subtle changes in performance across a large range of stimulus contrasts, we opted to acutely and reversibly silence VC using an optogenetic approach, thereby obtaining trials with and without VC silencing within the same session. Using this approach, we compared the impact of VC silencing on behavior in the same animals before and after ablation of CT neurons (*Figure 5*). Trained animals were tested on stimuli presented at seven different contrasts (0%, 4%, 8%, 16%, 32%, 64%, 100%) to obtain a psychometric function of visual sensitivity. On a third of the trials, VC was silenced by optogenetically activating GABAergic neurons expressing Channelrhodopsin 2,

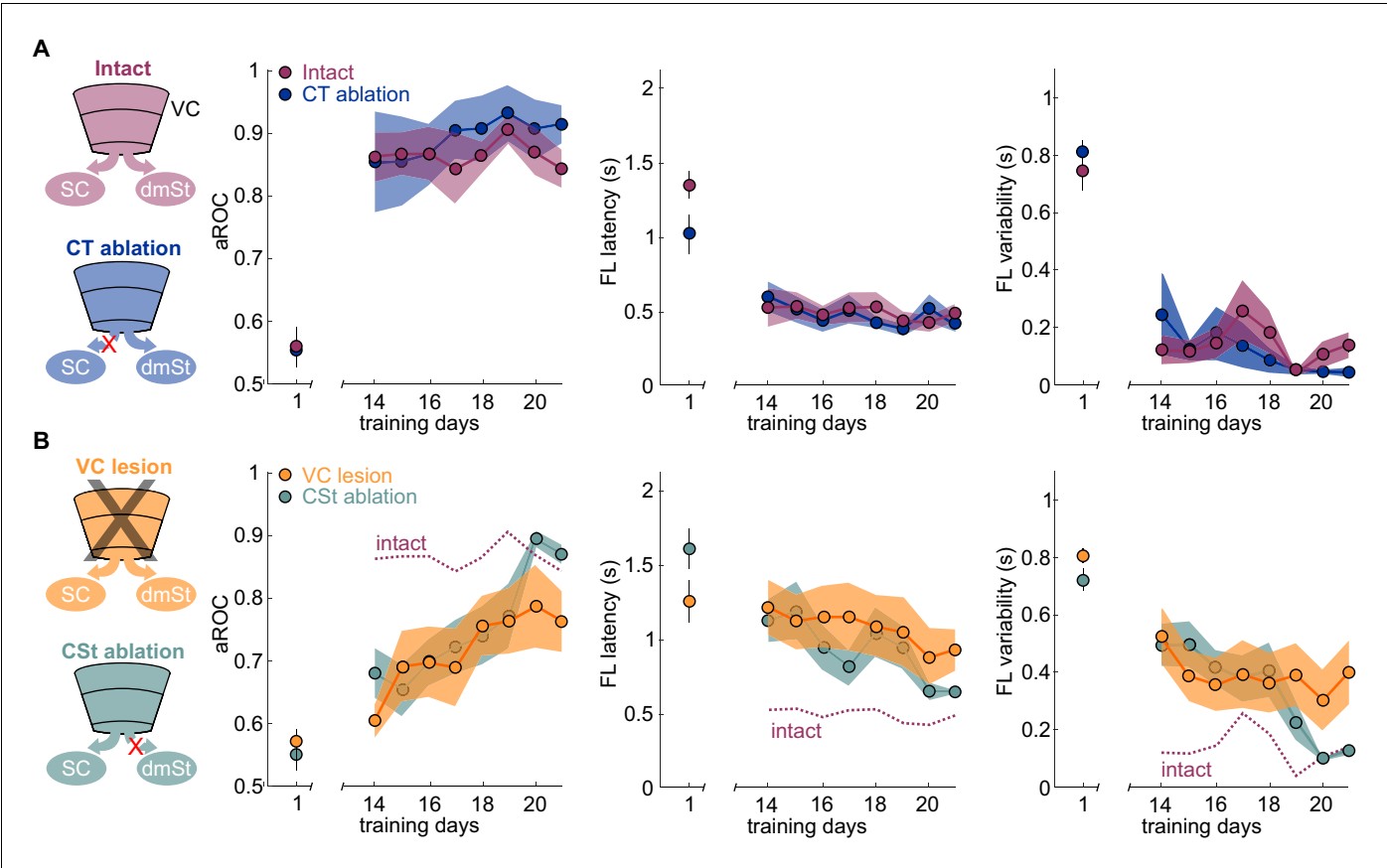

**Figure 3.** Ablation of cortico-striatal neurons reduces learning speed. (**A**) Left: Schematic of experimental groups. Right: Population average learning curves during the third week of training for intact (N = 8 mice) and CT-ablated mice (N = 5 mice). The first data point on the left of each graph is the value on the first day of training. aROC (left), FL latency for stimulus trials (middle) and FL variability for stimulus trials (right). Data plotted as mean ± SEM (N = 8 mice). Note that aROC, FL latency and variability have plateaued around day 14. (**B**) As above but for VC (N = 8 mice) and CSt-ablated (N = 8 mice) mice. The purple dotted line is data from intact animals (from A) for comparison. Note that over the third week of training, the aROC for both VC and CSt-ablated animals progressively approaches the performance levels of intact animals. By day 21, the aROC values are no longer significantly different across groups. The First lick latency and first lick variability also tend to decrease over the same period.

The online version of this article includes the following source data for figure 3:

**Source data 1.** Behavioral performance measurements throughout the third week training across experimental groups.

as described previously (*Lien and Scanziani, 2013*; *Olsen et al., 2012*). Consistent with the above results in which we surgically removed VC, optogenetic silencing of VC did not impair performance in response to stimuli presented at full contrast (aROC control: 0.88 ± 0.02 vs. aROC VC Silencing: 0.90 ± 0.04; mean ± SEM; N = 6 mice, n.s. Wilcoxon signed rank test; *Figure 5A*). However, at lower contrasts, silencing VC significantly reduced performance leading to a rightward shift of the psychometric function and a corresponding increase in the contrast threshold for detection (threshold contrast: control 7.8 ± 1% vs. VC silencing: 28 ± 3%, Weibull fit; see Materials and methods, p<0.01 Wilcoxon signed rank test; *Figure 5A*). Thus, VC modulates sensitivity by lowering the contrast threshold for stimulus detection. To determine whether VC modulates detection sensitivity via cortico-fugal neurons that target the SC, we ablated CT neurons using the same intersectional strategy described above. We noted that the animals did not exhibit the same level of performance at maximal stimulus contrast under control conditions before and after CT-ablation (contrast 100%: control pre CT-ablation aROC 0.88 ± 0.01 vs. control post CT-ablation aROC 0.83 ± 0.02; N = 3 mice; p<0.05; Mann-Whitney U test; *Figure 5B*), possibly because of the 3-week long training gap between the two conditions. Therefore, we assessed the role of CT-ablation by comparing the relative impact of VC silencing before and after the CT-ablation on the contrast threshold within the same test session. Strikingly, upon ablation of CT neurons the acute silencing of VC no longer lead

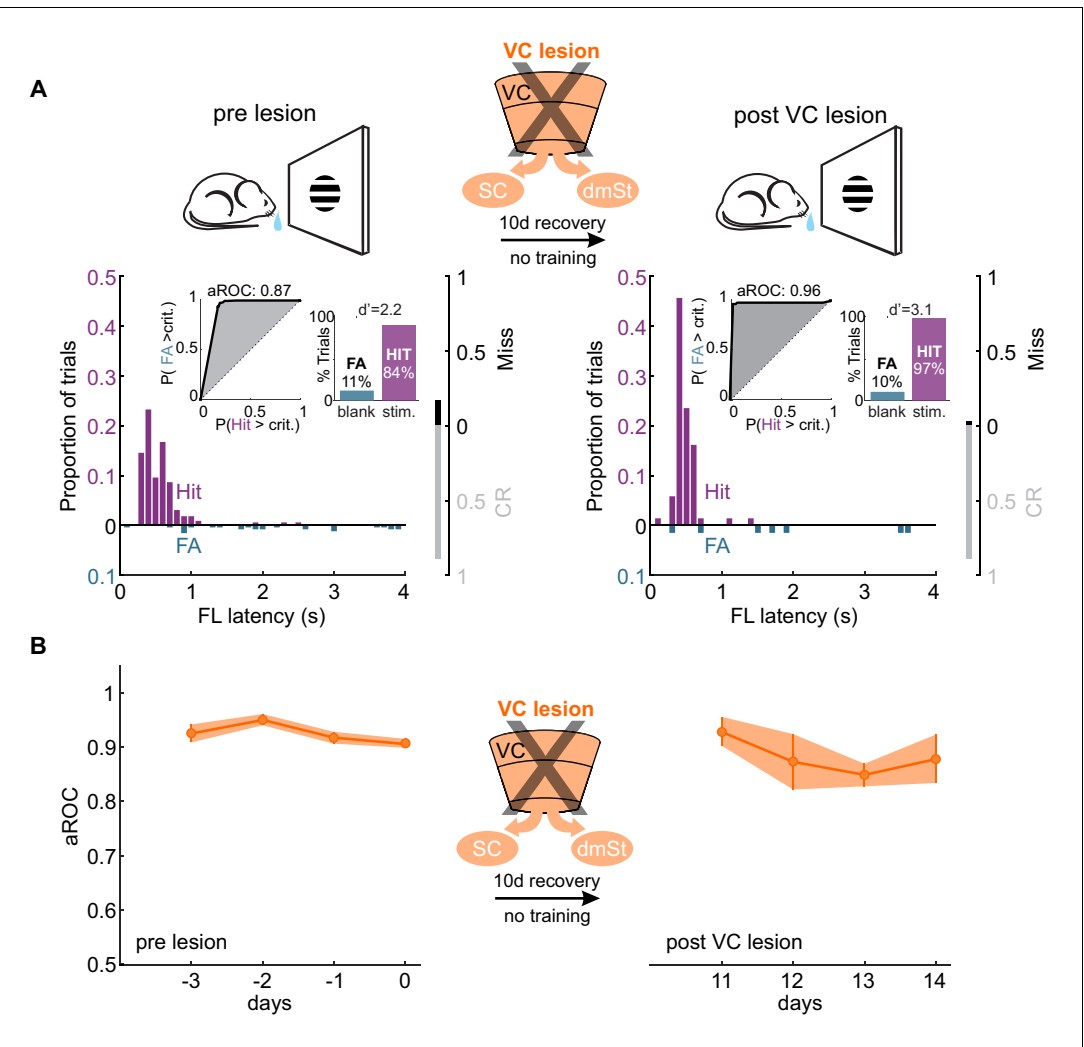

**Figure 4.** VC lesion after task acquisition does not impair task execution. (**A**) Top: Schematic of experimental design. After assessing the visual specificity of proficient mice, VC was removed surgically and their performance reassessed ten days later with no training in between. Bottom: Data from example mouse (illustrated as in *Figure 1B*) following completion of training (left) and 10 days after VC lesion (right). Inset: aROC curve for Hit versus FA trials and probability of Hits and FA and d-prime for the maximal response window. (**B**) Population average aROC for four consecutive sessions before (left) and 10 days after VC lesion (N = 4 mice). Data plotted as mean ± SEM. Note that the performance of trained animals following VC lesion is similar to that before lesion. The online version of this article includes the following source data and figure supplement(s) for figure 4:

**Source data 1.** Behavioral performance measurements as a function of VC lesion.
**Figure supplement 1.** VC lesion after task acquisition does not impair task execution assessed as visual specificity, first lick latency and variability of first lick latency.
**Figure supplement 1—source data 1.** Behavioral performance measurements as a function of VC lesion.

to an increase in the detection threshold (pre CT-ablation: threshold contrast: control 13.9 ± 3.3% vs. VC silencing: 35.4 ± 5.4%, p<0.05; post CT-ablation: threshold contrast: control 25.8 ± 9.5% vs. VC silencing: 31.2 ± 15.8%, N = 3 mice; n.s. Wilcoxon signed rank test; *Figure 5B*). These data indicate that cortico-fugal neurons projecting to the SC, while not involved in learning, increase detection sensitivity and thus enhances an animal's ability to detect less salient stimuli.

Do CT neurons maintain their role on detection sensitivity or, like the CSt neurons, eventually become dispensable for performance? To address this question, we compared the impact of VC silencing on detection sensitivity as a function of training duration. Interestingly, while animals trained 2–7 weeks (31 ± 12 days median ± SD; N = 6 mice) showed a strong rightward shift in the

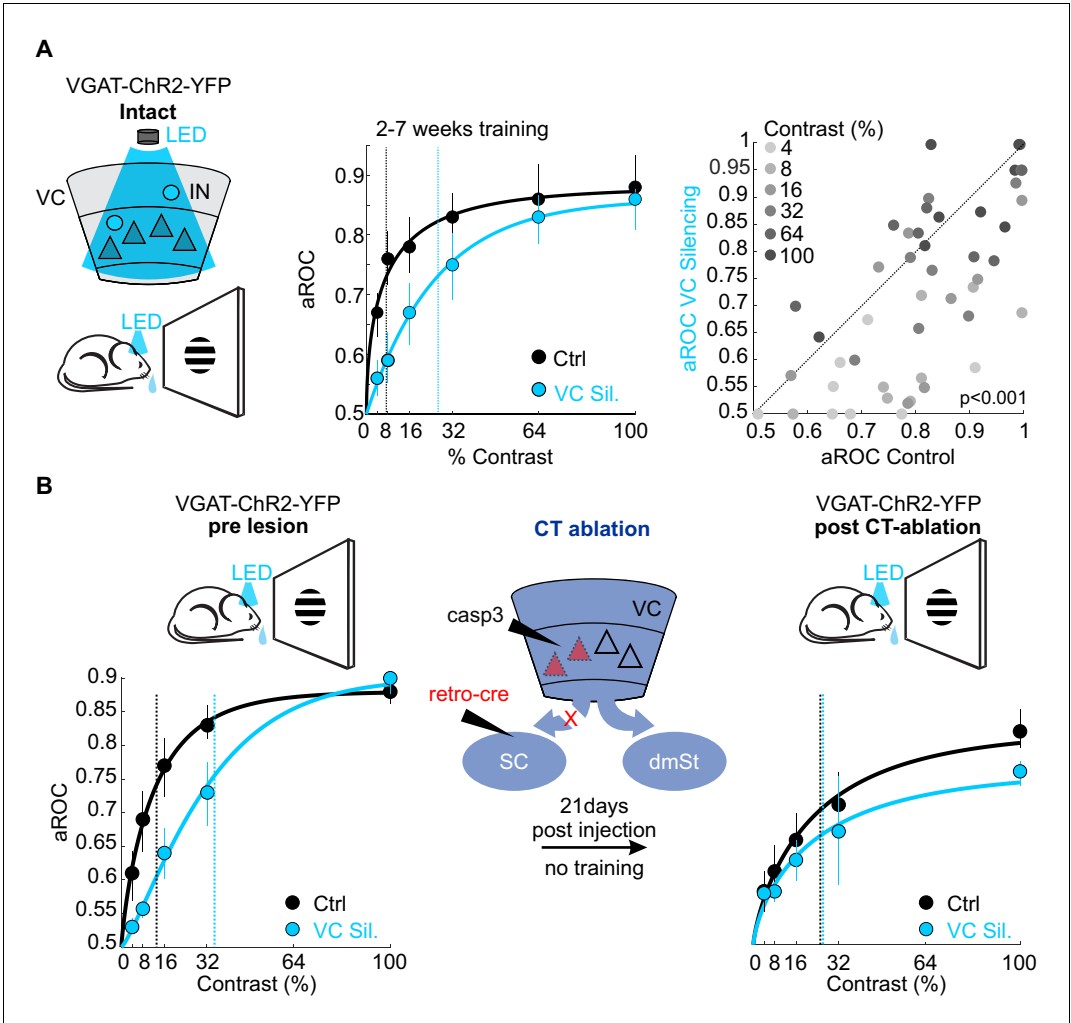

**Figure 5.** Ablation of cortico-tectal neurons increases the detection threshold. (**A**) Left: Schematic illustration of optogenetic silencing of VC by photo-activating inhibitory neurons (IN) in a behaving VGAT-ChR2-YFP animal. Middle: Psychometric function of aROC against stimulus contrast (0%, 4%, 8%, 16%, 32%, 64%, 100%) under control conditions (black) or during VC silencing (light blue; N = 6 mice, p<0.001 for contrast 4–32%, Mann-Whitney U test). Data plotted as mean ± SEM. Psychometric curve fits based on Weibull function. Dashed lines: contrast detection threshold. Note rightward shift of detection threshold upon VC silencing. Right: Scatter plot of aROC in control conditions versus VC silencing. Each dot represents aROC at a specific contrast within a behavioral test session (p<0.001 Wilcoxon signed-rank test; contrasts color coded from light gray to dark gray (4% to 100%, six contrasts), N = 6 mice). Note the stronger impact of VC silencing on aROC for lower contrast stimuli. (**B**) Left: Top schematic illustration of acute optogenetic silencing of VC in a mouse before CT ablation. Bottom: Psychometric function of aROC against stimulus contrast of trained mice under control conditions (black) or during VC silencing (light blue) before CT ablation (data plotted mean ± SEM; N = 3 mice; p<0.01 Wilcoxon signed-rank test). Middle: Schematic illustration of CT ablation. Right: Psychometric function of the same animals shown on the left 3 weeks after CT lesion without training in between. Dashed lines indicate the contrast detection threshold. Note that, in CT-ablated mice, VC silencing leads to almost no rightward shift of the contrast threshold.
The online version of this article includes the following source data for figure 5:

**Source data 1.** Behavioral performance measurements as a function of optogenetic silencing of VC and CT ablation.

psychometric function and an increase in detection threshold of approximately 3.4-fold upon VC silencing (threshold contrast: control 7.75 ± 1.8%, VC silencing: 26.6 ± 5.0%, p<0.01; Wilcoxon signed rank test; N = 6 mice; *Figure 6A*), detection sensitivity of animals trained 10–17 weeks (93 ± 9 days, median ± SD; N = 4 mice) was much less affected by VC silencing as the detection threshold upon VC silencing increased only 1.5 times (threshold contrast: control 10.02 ± 2.1%, VC

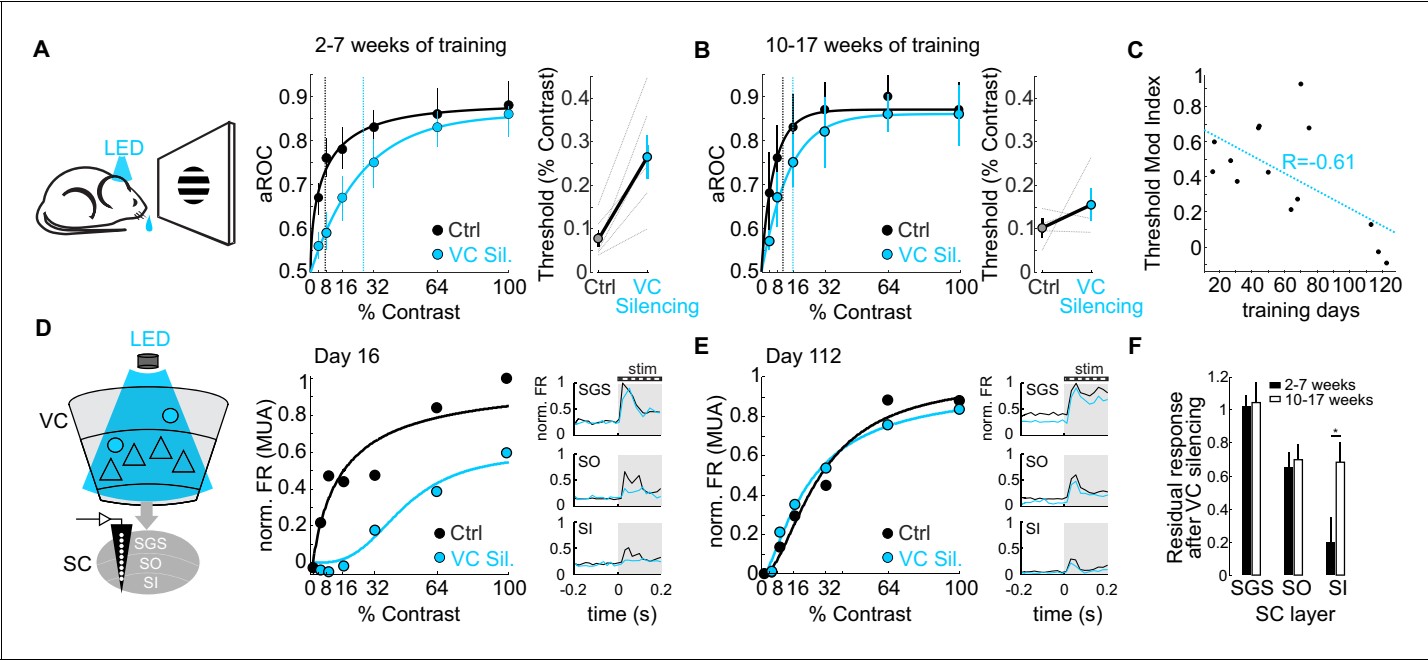

**Figure 6.** The impact of the cortico-tectal neurons diminishes with training. (**A**) Left: Schematic illustration of acute optogenetic silencing of VC in a behaving animal. Middle: Psychometric function of aROC against stimulus contrast of mice trained between 16 and 45 days under control conditions (black) and VC silencing (light blue; N = 6 mice). Data plotted as mean ± SEM. Dashed lines indicate the contrast detection threshold based on the Weibull function fit. Right: Contrast detection threshold of individual mice under control and during VC silencing (data plotted as mean ± SEM, N = 6 mice). (**B**) Left: As in A but for a separate group of mice trained between 70 and 131 days. Data plotted as mean ± SEM. Right: Contrast detection threshold of individual mice under control and during VC silencing (data plotted as mean ± SEM, N = 4 mice). Note that in these animals, VC silencing leads to a smaller rightward shift of the contrast detection threshold as compared to the animals with less training shown in (**A**). (**C**) Change in detection threshold upon VC silencing (reported as modulation index; see Materials and methods) plotted against training days. Each dot represents an individual animal. Dotted line: linear fit. Note inverse correlation between change in detection threshold and training days. (**D**) Left: Schematic illustration of extracellular recording in SC during behavior upon silencing of VC (stratum griseum superficiale (SGS); stratum opticum (SO); stratum griseum intermediale (SI)). Middle: Example mouse after 16 days of training. Contrast response function of multi-unit activity (MUA) in visual layers of SC under control conditions (black) and during VC silencing (light blue) during the performance of the detection task. Right: Peristimulus time histogram (PSTH) of MUA across depth in SC under control conditions (black) and during VC silencing (light blue; top: SGS; middle: SO; bottom: SO-SI). The shaded area is the period of stimulus presentation. Note the stronger effect of VC silencing on SI activity as compared to SGS. (**E**) As in (**D**) but for an example mouse after 112 days of training. (**F**) Population average of normalized evoked activity as a function of depth in SC. Black: early group (2–7 weeks of training), White: late group (10–17 weeks of training). Data plotted as mean ± SEM (early: N = 6 mice, late: N = 4 mice; p<0.05 Wilcoxon signed-rank test). Note that with prolonged training the impact of VC silencing on SI activity is strongly reduced.

The online version of this article includes the following source data for figure 6:

**Source data 1.** Measurements of the cortical impact on psychometric data as a function of prolonged training and on neuronal activity in the Superior Colliculus.

silencing.: 15.5 ± 3.7%, n.s.; Wilcoxon signed rank test; *Figure 6B*). Thus, the impact of VC on detection sensitivity diminishes with training (Pearson correlation coefficient R = −0.612; p=0.013; $R^2$ = 0.37; N = 16 mice; *Figure 6C*).

Given that VC modulates detection sensitivity via CT neurons (see above), we tested the impact of VC on visual responses in SC as a function of training duration. To this end, we recorded neuronal activity across the depth of SC using linear extracellular probes as the animal's performed the task. Early in training, VC silencing robustly reduced visual responses in the cortical recipient layers of SC, namely the stratum opticum and stratum griseum intermediale, and less so in the retino-recipient layer (*Figure 6D*). Later in training, visual responses across SC layers were less dependent on VC input (*Figure 6E,F*). Thus, the impact of the CT neurons both on detection sensitivity and on visual-evoked activity in the SC diminishes progressively with prolonged training.

## Discussion

The mammalian sensory cortex is an important stage in the representation of sensory stimuli, yet its contribution to the learning and performance in goal-directed behaviors is still controversial and may depend on whether the task involves detection or discrimination, and on the sensory modality required for the task (*Guo et al., 2014*; *Hutson and Masterton, 1986*; *Lashley, 1931*; *Miyashita and Feldman, 2013*; *Talwar et al., 2001*). Our results show that VC contributes to two select behavioral components of a simple visual detection task through two distinct populations of cortico-fugal neurons. On the one hand, VC, via the CSt neurons, plays a crucial role in the speed at which an animal learns to report the presence of visual stimuli. On the other hand, once the task has been learned VC modulates detection sensitivity via the CT neurons. Thus, this study reveals the functional dissociation of two major populations of cortico-fugal neurons in VC during specific stages in the learning and performance of a simple goal-directed behavior. Furthermore, eventually, both populations of cortico-fugal neurons become dispensable for the task.

Neurons projecting to the dmSt play their main role during the acquisition phase of the detection task as the ablation of CSt neurons impairs the learning speed but not the animal's ability to detect salient visual stimuli once the task has been learned. This is consistent with the role of the dmSt in action selection/initiation and reinforcement learning of goal-directed behavior (*Cox and Witten, 2019*; *Klaus et al., 2019*). The finding that VC via CSt neurons plays a preferential role during learning is reminiscent of the necessity of motor cortex in the learning but not for the performance of some motor tasks (*Kawai et al., 2015*). Given that the elimination of CSt neurons once the task has been learned does not affect performance, CSt neurons may function as a tutor to mediate learning-related plasticity in subcortical circuits that underlie the performance of the task. It will be interesting to address which properties of the subcortical circuitry that are not dependent on the CSt pathway once learning is achieved, are affected by the CSt pathway during learning. Our data suggest that the plastic events that occur during learning, while relying on the CSt input, do not reside at the CSt synapse. Otherwise removing VC after learning would impair performance. Given the existence of multiple subcortical loops between the SC and the basal ganglia (*Chevalier and Deniau, 1984*; *Harting et al., 2001*; *Krauzlis et al., 2013*; *Krout et al., 2001*; *Lin et al., 1984*; *McHaffie et al., 2005*; *Takada et al., 1985*), one could speculate that CSt neurons modify a SC-basal ganglia loop during learning. We cannot exclude that the ablation of CSt neurons also affects detection sensitivity, similar to the ablation of CT neurons. However, this possibility seems unlikely because silencing the entire VC (thus also including CSt neurons) following the ablation of CT neurons has little effect on the psychometric function.

The ability of neurons in sensory cortex to encode specific spatial and temporal features of the stimulus may likely be the neural basis for the role of VC (*Glickfeld et al., 2013*; *Jurjut et al., 2017*; *Marques et al., 2018*; *Petruno et al., 2013*; *Poort et al., 2015*; *Resulaj et al., 2018*) and in particular of CSt neurons (*Xiong et al., 2015*; *Znamenskiy and Zador, 2013*) in feature discrimination tasks. Also for the learning of the detection task, like the one used in this study, CSt neurons may convey an instructive signal about the spatio-temporal properties of the visual stimulus. Alternatively, for a detection task, CSt neurons may simply provide a permissive, and possibly unspecific, drive to subcortical structures necessary for animal's ability to form an association between a stimulus, an action and the outcome.

Contrary to the ablation of CSt neurons, the ablation of neurons that project to the SC did not not impact learning, at least when assessing learning with stimuli at 100% contrast. We cannot exclude that if learning had been assessed with lower contrast stimuli, that ablation of CT neurons could have revealed impairments in other aspects of learning, like perceptual learning (*Hua et al., 2010*; *Lu et al., 2011*; *Yan et al., 2014*). Ablation of CT neurons, however, impaired detection sensitivity. This result is consistent with the impact of cortical silencing on the sensory threshold for visually guided innate behavior mediated by the SC (*Liang et al., 2015*) and the involvement of the SC in simple orienting behaviors including voluntary licking (*Rossi et al., 2016*). Thus, CT neurons may represent the neural substrate underlying reported impairments in the perception of low-contrast stimuli following VC lesions (*Cowey and Stoerig, 1995*; *Glickfeld et al., 2013*; *Pöppel et al., 1973*; *Weiskrantz et al., 1974*). In humans and primates, this phenomenon is called blindsight and, interestingly, relies on the SC (*Cowey and Stoerig, 1995*; *Kinoshita et al., 2019*; *Mohler and Wurtz, 1977*), the main target of CT neurons. We show that VC enhances neuronal activity in SC in

response to visual stimuli, which may account, at least in part, for the deficits observed following VC lesions. With prolonged training, VC gradually loses its impact on behavior and on visual responses in SC, suggesting the existence of plasticity at the CT synapse or local plasticity within the SC.

The ablation of a specific population of cortico-fugal neurons may have effects on the connectivity or function of the remaining network in the VC. However, given that cortical neurons with distinct projection targets have been shown to form segregated subnetworks (*Brown and Hestrin, 2009*; *Harris and Mrsic-Flogel, 2013*; *Kim et al., 2018*; *Lur et al., 2016*; *Zhang et al., 2016*), we believe that the connectivity and function of the spared neurons is largely unaffected. In the future, optogenetic or pharmacogenetic approaches to transiently perturb the activity of cortico-fugal neurons may function as a complementary approach to mitigate any putative long-term compensatory changes in cortex. Ironically, however, even the impact of acute perturbations on behavior can be difficult to interpret specifically because of the lack of compensation in the downstream targets (*Otchy et al., 2015*).

Which structures may mediate learning, albeit slow, in the absence of VC? Given that, following VC lesions, the SC is the main visual processing stage in the brain and given the role of the cortico-fugal pathway targeting the dmSt in learning, we hypothesize that a subcortical SC-basal ganglia loop mediates the slower, VC-independent learning. Along these lines, recent findings indicate that a somatosensory detection task can be learned to proficiency in the absence of somatosensory cortex (*Hong et al., 2018*), possibly involving subcortical basal ganglia loops (*Bosman et al., 2011*; *Redgrave et al., 2010*). The fact that, with prolonged training, the behavior becomes independent of VC implies that the underlying subcortical structures can entirely rely on ascending sensory input from the periphery rather than on descending cortico-fugal pathways.

The gradual reduction in the role of VC on behavior may, furthermore, reflect a transition from goal-directed to habitual behavior (*Balleine, 2019*; *Dickinson, 1994*). On one hand, VC, through its cortico-fugal projection to the dmSt, may facilitate behavioral flexibility in an ever-changing environment during goal-directed behavior. On the other hand, the subcortical sources of visual input to the basal ganglia may be sufficient to drive habitual responses upon prolonged training in stable contingencies.

In summary, two distinct populations of cortico-fugal neurons in the VC that give rise to major descending projections targeting phylogenetically older subcortical structures, substantially contribute to visually guided behavior of a simple detection task, by increasing learning speed and improving detection sensitivity. These results establish a causal relationship between anatomically defined classes of cortico-fugal neurons and the emergence of a visually guided behavior and highlight the selective role of distinct classes of cortico-fugal neurons in specific aspects of simple goal-directed behavior. The improvement in learning and performance by defined classes of cortico-fugal neurons may represent an early adaptive benefit to goal-directed behavior associated with the expansion of cortex in mammals.

# Materials and methods

### Key resources table

| Reagent type (species) or resource | Designation | Source or reference | Identifiers | Additional information |
|---|---|---|---|---|
| Genetic reagent (*Mus musculus*) | VGat-ChR2-EYFP | Jackson Labs PMID:21985008 | RRID:MGI:4950481 | Dr. Guoping Feng (Massachusetts Institute of Technology) |
| Genetic reagent (*M. musculus*) | Ai14 | Jackson Labs PMID:20023653 | MGI:J:155793 | Dr. Hongkui Zeng (Allen Institute for Brain Science) |
| Genetic reagent (*M. musculus*) | Rosa26-LSL H2B-mCherry | Jackson Labs PMID:25913859 | MGI:J:221246 | Dr. Karel Svoboda (Janelia Farm Research Campus) |
| Antibody | Anti-NeuN (Rabbit polyclonal) | Abcam | Ab104225 RRID:AB_10711153 | IHC 1:1000 |
| Antibody | Alexa Fluor 594 (Goat Anti Rabbit IgG) | Thermo Fisher | A-11012 RRID:AB_2534079 | IHC 1:1000 |

*Continued on next page*

*Continued*

| Reagent type (species) or resource | Designation | Source or reference | Identifiers | Additional information |
|---|---|---|---|---|
| Peptide, recombinant protein | Cholera Toxin B (Alexa Fluor 488) | Thermo Fisher | C34775 | 1.0 mg/mL |
| Peptide, recombinant protein | Cholera Toxin B (Alexa Fluor 647) | Thermo Fisher | C34778 | 1.0 mg/mL |
| Recombinant DNA reagent | retroAAV-Cre; pm Syn1-EBFP-Cre | Addgene | RRID:Addgene_51507 | Dr. Hongkui Zeng (Allen Institute for Brain Science) |
| Sequence-based reagent | AAV2retro | Addgene | RRID:Addgene_81070 | Dr. Alla Karpova (Janelia Farm Research Campus) |
| Peptide, recombinant protein | taCasp3; Casp3 | Addgene | 45580 | Dr. Nirao Shah (University of California San Francisco) |

## Transgenic mice

All experimental procedures were performed with the approval of the Committee on Animal Care at UCSD and UCSF. Mice were housed on a reverse light/dark cycle (12/12 hr) and experiments were performed during the dark cycle. All animals were male and older than 8 weeks at the start of experiments. While on water restriction mice were single-housed and received a running wheel plus shelter for environmental enrichment. The mice used in this study were kept on a C57BL/6 background and were of the following genotype: VGat-ChR2-EYFP mice (Jackson Laboratories; stock #014548) which express Channelrhodopsin2 in gabaergic interneurons and VGat-ChR2-EYFP x ROSA-LSL-tdTom (Jackson Laboratories; stock# 007914) which express tdTomato after excision of a stop cassette by Cre recombinase and Rosa26-LSL-H2B-mCherry mice (Jackson Laboratories; stock #023139). Optogenetic experiments used heterozygous mice for the VGat-ChR2-EYFP transgene. All data on the time course of learning were acquired in male F1 offspring of VGat-ChR2-EYFP crossed with ROSA-LSL-tdTom reporter mice.

## Surgery and viral injections

Mice were anesthetized with 1.5–2% isoflurane and placed in a stereotactic apparatus (Kopf). The body temperature was measured using a rectal probe and maintained at 37°C with a heating pad (FHC; DC Temperature Controller). The eyes were protected by a thin layer of eye ointment (Rugby Laboratories) throughout the surgery. The animal's fur on the top of the head was shaved and the skin disinfected with Betadine. Topical lidocaine cream (2%, Akorn Pharmaceuticals) was administered at the incision site and the animals received a subcutaneous injection of 0.1 mg/kg Buprenorphine as postoperative analgesic.

## Cortical ablation

Animals were anesthetized as described above. The skull above VC was marked by using stereotaxic coordinates from Paxinos and Franklin mouse brain atlas (*Paxinos and Franklin, 2008*) and the skull at the marked area was thinned with a dental drill (700–900 μm). A drop of sterile phosphate buffer saline (PBS) was added to protect the exposed area before removal of the bone. The dura was removed and a cut of 1 mm depth was performed around the outline of VC using a microsurgical blade (FST 10316–14). The cortical tissue was removed (contralateral VC-lesion: five mice, bilateral VC-lesion: three mice) using a spoon shaped microsurgical blade (FST 10317–14) and the ablated area was washed with sterile PBS and presoaked Surgifoam to remove blood. Subsequently, the ablated area was protected by a layer of Silicon Kwik-Cast (WPI) followed by a layer of cyanoacrylate glue. Finally, dental cement was applied to permanently cover the lesions site.

## Viral injections

Using a dental drill, a craniotomy (about 100 μm diameter) was made over the injection site. Viral solutions were loaded in glass capillaries (beveled tip diameter 20–40 μm) and injected via a micropump (UMP-3, WPI) at a rate of 30 nl/min. The following adeno-associated viruses (AAV) were used: AAV2retro pmSyn1-EBFP-Cre (retroAAV-cre; titer: $5 \times 10^{12}$ vg/ml; Addgene virus catalog #51507-

AAVrg) and AAV2.1 flex-taCasp3-TEVp (AAV-Casp3; titer: 2.1 × 10$^{12}$ vg/ml; by Shah N., UNC). To label corticotectal (CT) or cortico-striatal (CSt) neurons, we unilaterally injected retroAAV-Cre in the right superior colliculus (SC; coordinates: 200 µm anterior and 700–800 µm lateral from lambda, three depths: 1500, 1400, 1300 µm below pia, 100–120 nl per site) or right dorso-medial striatum (dmSt; coordinates: anterior-posterior 0.9 mm and 1.5 mm lateral from bregma, 2.0 mm below pia, 300 nl), respectively. For cortical injections, we injected 3–4 sites (100–150 nl per site) of the right primary VC forming a triangle to target V1 (2.3 mm medio-lateral, 0.45 mm anterior from lambda; 2.8 mm medio-lateral, 0.45 mm anterior from lambda and 2.5 mm medio-lateral, 1 mm anterior from lambda) with AAV-Casp3. The pipette was removed approximately 15 min after the injection was completed, to prevent leakage of the virus along the injection tract. The skin was sutured with suture silk (Fisher Scientific NC9134710).

## Headbar implantation

Each animal was implanted with a custom-made headbar for head fixation. Briefly, animals were prepared as described above. Upon exposure of the skull by removal of the skin and periosteum, the bone was cleaned with a sterile cotton swab and sealed with Vetbond (Fisher Scientific). The headbar was fixed with a layer of cyanoacrylate glue followed by a layer of black dental cement (Lang Dental; Ortho-Jet BCA) to ensure long-term affixation of the headbar. The dental cement was used to build a recording well around the area of the right VC and SC, which was protected by a thin layer of transparent cyanoacrylate glue to permit access for light stimulation.

## Craniotomy

On the day before the electrophysiological recordings, animals were anesthetized with 1.5% isoflurane and a craniotomy was made over SC (diameter: ~400 µm, anterior-posterior 200 µm medio-lateral 700–800 µm from lambdoid suture) or V1 (diameter: ~400 µm, anterior-posterior 200 µm medio-lateral 2.3 mm from lambdoid suture). The craniotomy was protected by a local application of Kwik-Cast (WPI) until the day of the recording.

## Behavioral setup

A schematic of the behavior setup is shown in *Figure 1*. Briefly, mice were head-fixed and crouched in a natural position in an acrylic tube with the paws resting on the edge of the tube. Water rewards were delivered by a custom-made optical lickometer that registered the movement of the animal's tongue. The lickometer was activated by interruptions of the light path between an LED and a phototransistor upon licking. The behavioral task was controlled by software (Rpbox; http://brodylab.org) running in MATLAB (MathWorks) communicating with a real-time system (RTLinux). Water was delivered by gravitational flow under the control of a solenoid valve (NResearch; Model 161K011; valve driver: CoolDrive) that was connected to the lick spout (hypodermic tubing; gauge 14) via Tygon tubing (1/16 inch ID) and calibrated the reward deliver to a ~ 3 µl drop of water per trial.

## Visual stimulation

Visual stimuli were generated in Matlab with Psychtoolbox and custom written stimulus software based on StimGen (https://github.com/mscaudill/neuroGit; *Ruediger, 2020*) and Vstimcontroller (https://github.com/aresulaj/ResRueOlsSca18; *Olen, 2020*) and presented on a gamma corrected LCD monitor (DELL, mean luminance: 60 cd/m2, monitor refresh rate 60 Hz: dimensions: 47.5 × 30 cm; 1680 × 1050 pixels) which was positioned at a distance of 14 cm from the left eye (contralateral to the right VC). The position of the monitor relative to the animal was angled at 45° from the long body axis.

For the experiments illustrated in *Figures 1–3* circular sinusoidal drifting grating patches were displayed on the monitor (patch diameter: 30°, spatial frequency: 0.04 cycles/degree, temporal frequency: 2 Hz, contrast range: 0, 4, 8, 16, 32, 64, 100%, horizontal grating moving upward). The center of the patch was placed 60° to the left of the mouse's midline, hence far from the binocular zone. Given that the stimulus radius was 15°, this leaves a margin of ~20–25° between the nasal edge of the stimulus and the temporal edge of the left binocular zone ensuring the stimulation of the left monocular visual field only.

For the experiments illustrated in *Figures 5* and *6* full field drifting gratings were used (spatial frequency: 0.04 cycles/degree, temporal frequency: 2 Hz, contrast range: 0, 4, 8, 16, 32, 64, 100%, horizontal grating moving upward). The full-field stimulus was placed in the left visual hemifield, approximately 45° from the mouse's midline covering approximately 110° of visual space (0° and 110° in azimuth). Thus, a portion of the stimulus extended into the binocular zone of the left visual field, possibly enabling the animal to use V1 ipsilateral to the stimulus to perform the task. However, even animals with bilateral lesion of VC maintained the ability to highly specifically respond to the stimulus (full-field stimulus contrast 100% aROC 0.92 ± 0.026; mean ± SD; N = 2 mice), indicating that the behavior does not rely on the VC ipsilateral to the stimulus.

The timing of the visual stimuli was controlled by the Real-Time Linux State Machine. During behavioral training, each stimulus was presented for up to 4 s and followed by a gray screen (luminance: 60 cd/m$^2$) for several seconds (*Figure 1—figure supplement 1A–B*). Visual stimuli were presented in blocks, composed of randomly interleaved trials, one for each contrast. The duration of the gray screen varied from trial to trial and was depending on the training stage of the animal (*Figure 1—figure supplement 1C*).

## Behavior task

### Task learning

Experimenters were blind to whether animals were in the lesioned or control group. For experiments aimed at determining the time course of learning, mice were trained with a standardized training routine that was identical for all animals. Animals were allowed to recover from headbar implantation for at least 3 days before the start of water restriction (≥1 ml/day). Upon weight stabilization (target weight loss of 15%) the pre-training stage consisted of 1–3 days during which animals were habituated to head fixation and licking on the lick spout. Progression through this pre-training stage depended on the animal's weight stabilization and whether the animal exhibited a high motivational level indicated by its spontaneous licking frequency (>75% of all rewards collected within a session of 150 trials).

In the training stage, mice were rewarded for initiating a lick on stimulus trials during which a moving grating was presented on the center of the monitor (luminance contrast c = 100%, 64%, 32%) for up to 4 s. The first lick during the presentation of the stimulus triggered the reward delivery and the next trial started with a gray screen. On blank trials, the luminance of the monitor was maintained constant and no visual stimulus was presented (c = 0%). Licks on blank trials (false alarms) were not punished. However, licking throughout a 1.5 s window preceding the stimulus was punished with a timeout by reinitiating the trial. There were no rewards for correct rejections and no punishments for misses. The first lick latency was defined as the time from stimulus onset to the first interruption of the optical lickometer.

The temporal structure of a trial consisted of a fixed inter-trial-interval (ITI) and a randomized jitter period ($0 \leq t_{jitter} \leq$ jitter period; drawn from a uniform distribution on each trial) during which the screen was gray (*Figure 1—figure supplement 1*). We also implemented a delay period ($t_{delay}$250 ms) at the beginning of the stimulus presentation period during which licks were not rewarded. This delay period helped enforce the association of the visual stimulus with reward and gradually decreased in duration with training. Time intervals were changed as a function of training (*Figure 1—figure supplement 1C*). The number of trials was limited to 250 per training session. There was only one session per day. The amount of reward per session was calculated based on the animal's weight difference before and after the training session and mice were individually supplemented with additional water if needed in order to maintain a stable weight loss across training days. Learning was considered completed once animals reached a stable performance defined as >0.8 aROC for four consecutive sessions. Data from retroAAV-Cre only animals was pooled from animals either injected in the SC (n = 3 mice) or the dmSt (n = 4) as we wound no statistical difference in the learning ability of both groups across training (day 1–14: average aROC/day: retroAAV-Cre only in dmSt: 0.023 ± 0.007 vs. in SC: 0.031 ± 0.006; mean ± SEM; n.s. Mann-Whitney U test).

### Task performance

For mice that were trained to perform the task but were not used to determine the time course of learning (see above), training parameters were adjusted on an individual basis. Under these

conditions, mice were limited to perform up to 500 trials per day. Once the mice learned the detection task for high-contrast stimuli (32%, 64%, 100% and blanks), the contrast sensitivity was measured by expanding the contrast range to lower contrast stimuli (4%, 8%, 16%).

## Optogenetic inhibition

We used VGAT-ChR2-EYFP and double transgenic VGAT-ChR2-EYFP x Ai14 mice for optogenetic silencing. ChR2-expressing inhibitory interneurons were activated via an optical fiber (diameter 1 mm) coupled to a blue LED (470 nm; Thorlabs) that was placed over VC. To test the behavioral effect of VC silencing the LED fiber was positioned approximately 3 mm above the thinned skull covered with a thin layer of cyanoacrylate glue and immersed in PBS. For optogenetic silencing during extracellular recording in SC, the fiber was also positioned above VC but immersed in ACSF throughout the recording. Light from the fiber optic covered the entire surface area of VC (Power: 10 mW measured at the fiber tip). The LED power was determined before each experiment using a power meter (Thorlabs, PM100D). Optogenetic silencing of neural activity in VC was highly efficient as demonstrated previously (*Lien and Scanziani, 2013*; *Olsen et al., 2012*). The onset and offset of the LED were controlled by the Real-Time Linux State Machine. The LED was turned on before the presentation of visual stimuli (or blanks) and lasted for the entire duration of the stimulus. An opaque shield of black insulation around the fiber was used to minimize the amount of blue light that may directly reach the animal's eyes. LED onset was randomized (0 to 1 s) in order to reduce the possibility that the onset of the LED may be used as a cue. A third of the trials were LED trials and were interleaved throughout the experiment.

## Electrophysiology

On the day of the recording, the animal was head-fixed and the protective layer of Silicon Kwik-Cast (WPI) was removed. The exposed tissue was kept moist with artificial cerebrospinal fluid (ACSF; 140 mM NaCl, 5 mM KCl, 10 mM d-glucose, 10 mM HEPES, 2 mM CaCl2, 2 mM MgSO4, pH 7.4). Extracellular recordings of neural activity were performed using linear silicon probes (Neuronexus, probe type: A1 × 16–5 mm-25-177, A1 × 32–5 mm-25-177, A1 × 32-Edge-5mm-20–177). The recording electrode was mounted on a micromanipulator (Luigs and Neumann) and stained with DiI (Life Technologies) for post hoc identification of the recording site and inserted in SC (coordinates: 200 µm anterior and 700–800 µm lateral to lambda). Signals were amplified and band pass filtered using 16-channels or 32-channels and A-M System headstages (gain 20x) connected to 16-channel and 32-channel A-M System amplifier, respectively (Model 3500 and Model 4000, gain 100x, band pass filter: between 0.3 Hz or 0.1 Hz and 5 KHz). The amplified signals were recorded at 32 KHz using a NIDAQ board (PCIe6259) controlled with custom-written software in Matlab (MathWorks). The tip of the electrode was lowered to a depth of approximately 1300–1500 µm from the pial surface to cover the visual layers of SC. Data collection started 20–30 min after probe insertion. For experiments during which we recorded neural activity in SC the duration of the response window was limited to 2 s. Recording depths across different experiments were aligned based on the normalized visual-evoked multi-unit activity across layers and the current-source density analysis (CSD) of the local field potential (*Stitt et al., 2013*; *Zhao et al., 2014*).

## Histology

Upon completion of behavioral testing, mice were perfused transcardially with 4% paraformaldehyde (PFA) in 0.1 M sodium phosphate buffer (PBS). The brains were extracted and post-fixed in 4% PFA overnight at 4°C, washed in PBS and sectioned coronally with a thickness of 60–100 µm thickness using a vibratome or microtome. For immunocytochemistry, sections were permeabilized with 0.2% Triton X-100 in PBS with 10% bovine serum albumin. Primary antibody incubation was overnight at 4°C. Secondary antibody incubation lasted 2–3 hr at room temperature. Slices were mounted using a Vectashield mounting medium containing DAPI (Vector Laboratories H1500). We verified the accurate targeting of the SC and of the dmSt through the local expression of tdTomato, in all experiments where we injected retroAAV-Cre virus, given that the Cre virus was injected in the Ai14 background. In cases where we used the retrograde tracer CTB, we validated the targeting using the local CTB labeling at the target site.

### Surgical lesions

Bright-field and fluorescence images were acquired on a macroView microscope (MVX10 Olympus) and the diameter of the surgical lesion was quantified based on the Feret diameter with ImageJ by outlining the lesioned area (cell reduction 90%) for a series of sections spanning VC and averaging across sections. All mice had lesions in V1 and surrounding higher visual areas that encompassed the part of the visual field which we used to present the visual stimuli and, in some cases, the lesions extended beyond VC into neighboring sensory areas such as somatosensory cortex.

### Quantification of retrograde tracing

A Nikon Ti CSU-W1 inverted spinning disk confocal microscope was used to acquire image stacks of Cholera Toxin B (CTB Alexa Fluor 488 Conjugate injected into dmSt, CTB Alexa Fluor 647 Conjugate injected into SC, Thermo Fisher) positive neurons in VC. To label all cortical neurons, we stained the sections with a NeuN antibody (1:1000 Abcam, ab104225). At least four sections were analyzed per animal, and the data are based on approximately 500–700 µm regions along the anterior–posterior axis. The cellular overlap of retrogradely labeled CSt and CT neurons was quantified based on single-cell analysis in z-stacks with ImageJ.

### Quantification of viral injections

To estimate the infection rate of the AAV2retro-pmSyn1-EBFP-Cre, we co-injected the virus with CTB in SC in a set of R26 LSL H2B mCherry animals. This allowed us to compare the number of CTB retrogradely labeled VC neurons with the number of nuclear mCherry-expressing cells. To estimate the effect of neuronal ablation upon taCasp3 expression in VC, we quantified the cell density in layer 5 using the neuronal marker NeuN.

## Data analysis

### First lick latency analysis

We computed the area under the receiver operating characteristic curve (aROC) for the temporal distribution of first lick latencies on stimulus and blank trials. To this end, the first lick latencies were binned (100 ms bins during the response window of 0–4 s). To include changes in the probability to initiate lick in the ROC analysis, we included Misses and Correct Rejections by assigning to these trials a time stamp outside of the response window (4.1 s).

### Detection threshold analysis

To determine detection threshold, we fitted the psychometric function (aROC against stimulus contrast) with the Weibull function using the Palamedes toolbox (**Kingdom and Prins, 2010**):

$$\Psi(x; \alpha, \beta, \gamma, \lambda) = \gamma + (1 - \gamma - \lambda) \, x \, Fw(x; \alpha, \beta)$$

$$Fw(x; \alpha, \beta) = 1 - e^{-(x-\alpha)\beta}$$

$\psi$ refers to the aROC as a function of stimulus contrast $x$. The lower asymptote of psi is given by $\gamma$ (aROC for 0% contrast), while the upper asymptote is determined by $\lambda$ (lapse rate), that is 1- (aROC for 100% contrast). $\alpha$ and $\beta$ are detection threshold and slope, respectively. To quantify the difference in performance between control (LED off) and VC silencing (LED on) trials, we calculated the modulation index (MI) of the detection threshold:

$$MI = \frac{LED\ on - LED\ off}{LED\ off + LED\ on}$$

### Trial-to-trial variability

The trial-to-trial variability of licking was calculated as the ratio of the variance to the mean of first lick latencies on a given training day per animal.

## Electrophysiology

### Spike sorting

Spikes were defined as events crossing a threshold of 4fourtimes the standard deviation of the amplified and high pass filtered (500 Hz) extracellular signal. Spike waveforms of 4fouradjacent electrode sites (trode) were sorted and clustered using the spike sorting software UltraMegaSort (https://physics.ucsd.edu/neurophysics). Clusters were manually classified into putative isolated units or multiunit activity based on their average waveform shapes. Criteria for isolated units were refractory period violations < 0.1%; fraction of spikes with amplitude below detection threshold (estimated by a Gaussian fit to the spike amplitudes distribution)<15%. To satisfy these criteria, outliers (such as noise or overlapping spike waveforms) were manually removed based on the distribution of the Mahalanobis distance of spike waveforms from the cluster center. Data from isolated and non-isolated units was merged to represent the overall multi-unit activity at a given trode.

### Contrast response function

Baseline subtracted multiunit activity (first 100 ms of visual stimulus presentation) was normalized and fitted with a hyperbolic ratio function:

$$r = \frac{r_{max} * x^n}{x^n + x_{50}^n}$$

$r$ refers to response as a function of the stimulus contrast $x$, $r_{max}$ is a fitted constant representing the saturation level of the response, n is the fitting exponent which affects the shape of the curve fit and $x_{50}$ corresponds to the semi-saturation constant.

## Statistical analysis

All data were analyzed with custom written Matlab code (MathWorks). Statistical tests were used as stated in the figure legends. All data are presented as mean ± standard error of the mean (SEM) across mice unless otherwise noted. In cases where we performed repeated measurements in single animals (e.g. psychometric performance measurements in *Figure 5* and *Figure 6*), we used the average performance of an individual across sessions to determine the statistical significance across independent samples. Statistical significance was determined using two-sided hypothesis tests and statistical significance is denoted as *p<0.05, **p<0.01, ***p<0.001.

## Acknowledgements

We thank all the members of the Scanziani lab for discussions about the project and comments on the manuscript; Alexandra Nelson and Jun Lee for critical reading of the manuscript; J Evora, N Kim, Y Li, L Bao, M Mukundan and B Wong for technical support.; F Alagala, D Brumby, M Calvert, A Chen, J Chen, L Chun, C Conroy, S Dennis, S Feng, P Gu, V Hovsepian, EK Gines, A Kaplan, L Ko, A Liu, D Major, P Reich, K Sanchez, A Singh, N Skajaa, D Unwalla, C Wang and J Zaragoza for help training animals. This project was supported by the European Molecular Biology Organization (postdoctoral long-term fellowship ALTF741-2012), The Swiss National Science Foundation (postdoc fellowships: 151168 and 138719), NIH R01EY025668 and the Howard Hughes Medical Institute.

## Additional information

### Funding

| Funder | Grant reference number | Author |
| --- | --- | --- |
| NIH | R01EY025668 | Massimo Scanziani |
| Howard Hughes Medical Institute | | Massimo Scanziani |
| European Molecular Biology Organization | ALTF741-2012 | Sarah Ruediger |
| Swiss National Science Foundation | 151168 | Sarah Ruediger |

| Swiss National Science Foundation | 138719 | Sarah Ruediger |
|---|---|---|

The funders had no role in study design, data collection and interpretation, or the decision to submit the work for publication.

## Author contributions

Sarah Ruediger, Conceptualization, Data curation, Software, Formal analysis, Funding acquisition, Validation, Investigation, Visualization, Methodology, Writing - original draft; Massimo Scanziani, Conceptualization, Supervision, Funding acquisition, Visualization, Writing - original draft, Project administration

## Author ORCIDs

Sarah Ruediger ![ORCID] https://orcid.org/0000-0001-6684-4693
Massimo Scanziani ![ORCID] https://orcid.org/0000-0002-5331-9686

## Ethics

Animal experimentation: All experimental procedures were performed with the approval of the Committee on Animal Care at UCSD and UCSF. Authorization # AN179056.

## Decision letter and Author response

Decision letter https://doi.org/10.7554/eLife.59247.sa1
Author response https://doi.org/10.7554/eLife.59247.sa2

# Additional files

## Supplementary files
• Transparent reporting form

## Data availability

All data generated or analyzed during this study are included in the manuscript and supporting files. Numerical data for graphs represented in figures 1-6, figure 1-figure supplement 2,3,4,5, figure 2-figure supplement 1, figure 4-figure supplement 1 are provided as source data files. The software used to generate visual stimuli and record neuronal activity is available at: https://github.com/mscaudill/neuroGit and https://github.com/aresulaj/ResRueOlsSca18.

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
