## [Decision Letter]

**Acceptance summary:**

This work uncovers the role of two distinct cortico-fugal pathways in the learning and the performance of a visual detection task. It demonstrates that visual cortex neurons that project to the striatum enhance learning speed, while visual cortex neurons that project to the superior colliculus enhance detection sensitivity. This study contributes to our understanding of the function of visual cortex during the learning and execution of a visual task.

**Decision letter after peer review:**

Thank you for submitting your article "Distinct cortico-fugal neurons in visual cortex enhance learning speed and detection sensitivity" for consideration by *eLife*. Your article has been reviewed by three peer reviewers, and the evaluation has been overseen by a Reviewing Editor and Andrew King as the Senior Editor.

The reviewers have discussed the reviews with one another and the Reviewing Editor has drafted this decision to help you prepare a revised submission.

Summary:

In this manuscript the authors investigate the different roles of two different subclasses of projection neurons in visual cortex during a simple stimulus detection task. The functional role of each subclass is assessed by targeting them for selective ablation based on their projection target. One cell type, projecting to the striatum, is vital for acquisition of the task but is dispensable for performance after acquisition. The second cell type, projecting to the superior colliculus, is not required for the mice to learn the task, but has a modulatory role during performance – activity in these cells enhances behavioural sensitivity moment-to-moment.

The findings are important and should be of great interest to the behavioural neuroscience community. The results are presented clearly, logically and succinctly. The reviewers have some concerns which can be addressed with editorial changes, further analysis and/or discussion. Some comments call for more histology which is hopefully possible. Other comments could be addressed by new experiments (relating to 1P stimulation), but these experiments should be considered optional.

Essential revisions:

1) The core of the story is learning; task acquisition and task performance. It is therefore quite surprising that behavioural performance seems so poor – there is a very high false alarm rate even in learned animals (86% probability of lick in the absence of a stimulus, Figure 1 far right example. Note the example in Figure 4 is much better). Could the authors please discuss this issue? While this does not challenge the results in the paper, it does raise some concern over the “expertness” of the animals, i.e. how well they have actually learned. This should be mentioned as a caveat and discussed. What could the possible reasons for the high false alarm rate be? The stimulus duration is quite long at 4 seconds; perhaps with a shorter stimulus (and thus shorter blank period and/or shorter response window) the performance would improve.

2) Further to the previous comment, the authors write “learning was considered complete by 14 days” but it does not appear as though the FA rate has stopped dropping by this time, i.e. the animals are still learning. While the performance in this simple task is not exactly exemplary, it is clear the mice are trying; the concern is how do these results depend on actual task performance, or “expertness”?

3) Wording of main claim: The authors stress throughout the manuscript that they have shown how Str neurons "enhance learning speed" while CT neurons "improve detection sensitivity". Instead what they have shown is that lesioning Str neurons slows down learning and lesioning CT neurons decreases sensitivity. The reversal of interpretation currently used is not justified, since if you claim Str neurons enhance learning speed, it begs the question, enhanced compared to what? We can only make such relative statements from the known rate of learning in an intact animal as a reference, and the correct conclusion is thus an impairment on lesioning. "Impairs the rate of learning", "is required for normal learning rates", etc are alternatives that the authors could use instead.

Importantly, the current phrasing leads the reader to assume that the authors have performed a manipulation that actually speeds up learning compared to controls, which is what the Abstract currently implies. I would advise the authors change all phrasing of this kind throughout the entire manuscript, including title and Abstract.

4) Statistics: There are a number problems with the statistics throughout the paper which must be corrected in order to justify the claims.

– In Figure 1 it is unclear what statistics have been done to compare the learning rates across conditions. The text only mentions slope, SEM, and n.s. No further information is available in the Materials and methods. This needs to be clarified. What exact test has been done to determine that the slopes in two linear regressions are significantly different? Have the authors calculated the confidence intervals of the slope using bootstrapping? Or have they performed linear regression analysis and measured the significance of an interaction term between session number and lesion condition?

– A serious concern is that in Figure 1E, the cortico-striatal lesion condition has been statistically tested against the cortico-tectal, rather than control mice. This does not support the claims made either in text or in the figure, where the control mice plots are overlaid on the same panel.

– Further, lesion of full visual cortex are compared with striatal lesion and a non-significant difference is reported. Clearly the correct test is to compare VC lesion with the VC controls (which the authors have done when comparing spontaneous lick rates).

– Spontaneous lick rates have been compared in some, but not other cases. This measure is particularly important for the striatal lesion case (see below). Could the authors specify what time periods are included in “spontaneous” and provide comparisons of spontaneous lick rate consistently.

– In a number of cases throughout the manuscript, statistics have been performed on N sessions, which includes multiple sessions from the same animals (e.g. 9 sessions from 6 mice). This is incorrect since multiple sessions from the same mouse are not independent samples. While errors of this kind are unfortunately common in the field at the moment, it is important to avoid it in general, and in particular in this study given that behaviour is highly correlated within individuals. One way to deal with data of this nature is to perform tests with fixed and random effects (e.g. as described in Aarts et al. Nat Neurosci 2014). Another option is to average sessions for each animal.

– Figure 5B: A key claim of this study is that CT lesions reduce sensitivity. However, we only found a comparison of the effect of optogenetic silencing vs. no silencing, but no test between the non-silenced pre vs. post CT lesion. Although the figure legend says “Note reduction in rightward shift of the contrast threshold after CT lesion“ this is not actually compared. This seems like a key comparison to support the main claim of the study.

5) Gross behavioural changes: Measures of gross behavioural changes should be measured with and without lesions in order to rule out the role of these changes in learning rates and sensitivity curves. Importantly, non-stimulus period lick rate should be compared between controls and visual cortex/cortico-striatal lesions, since a reduction in overall lick rate may account for the lower rate of learning.

– The number of trials was limited to 250 per training session: Could the authors clarify if all mice always reached 250 trials in each session, and if not, report the average and range of trials actually performed by the mice in each group. With this information, could the authors rule out that any differences in rates of learning were due to different numbers of trials performed in some days by lesioned mice.

6) Latency of first licks on blank trials: In the behavioural paradigm used here, latency from blank trials onset is not really a latency, since the mouse experiences a continuous blank screen even though the software might have transitioned from the delay to the “blank” stimulus. This point should be made clear to the reader in the Results and Materials and methods sections, especially when presenting plots like Figure 1B where histograms of lick latencies on blank trials are presented.

7) The Introduction is heavily focused on cross species comparisons and evolutionary arguments, and appears more appropriate for a cross species comparative study. In particular, it seems to set up a comparison between mammals, with expanded cortico-fugal pathways, and non-mammals without this expansion. While this is an entirely subjective judgement, this study would benefit from an Introduction more suited to the questions addressed.

8) Throughout the manuscript, emphases such as the following occur along with essentially each result presented:

– "the visual specificity of the licking behavior increased much slower"

– "learning progressed much slower and.… was far from complete."

– "… strongly reduced the impact"

The authors should either substantiate what “much slower”, “far from complete” etc means, or rephrase as much as possible.

9) It's not immediately clear what the timing of the task is. The schematic in Figure 1A should be extended so as to indicate the timing of the task. The Materials and methods state the stimulus duration is "up to 4 seconds". Why up to? What about the time between trials? What was the actual ITI? Was the no-lick requirement invalidated often for these mice? (do the mice lick constantly or have they learnt to withhold the licking mostly but the 4 second blank is just too long in addition?)

10) The authors use a metric that is unconventional – aROC of lick latencies. This reaction time metric does make sense and appears convincing from the example in Figure 1. While I like the metric, it is not standard in the field. Therefore it would be nice to more fully compare it to other standard metrics for this type of behaviour – i.e. P(Lick) and d-prime. These comparisons are indirectly available, but It would be better to directly compare a sessions P(Lick) and/or dprime with that session's aROC.

11) The bulk of this paper makes use of selective ablation of neurons based on their projection target, via a combination of retroAAV-Cre in the target area and AAV-Caspase3 in the source area. There are no references to existing literature regarding this method: could the authors please add some? It would also strengthen the manuscript if authors could also demonstrate the specificity of this method in at least one of two ways:

a) First, how specific is the cell death in the source location? Could the authors provide histology showing that other neurons are indeed spared? What are the consequences of the local death of a good many cells? Is the function and connectivity of other cells unaffected? (perhaps the literature could provide a hint, these new experiments would be a considerable undertaking. At the least, the potential effects should be discussed). When the authors describe the “lesion” of CT and CSt, is it a lesion in the same sense as the VC lesions? Perhaps “ablation” is a better word and does more justice to the method. It would be good to see Caspase-only (no Cre) controls, in addition to the Cre-only controls. Does VC remain healthy? Note the concentrations of the virus used are not described in the Materials and methods.

b) Second, how accurate are the target site injections? It would be good practice to provide histology confirming accurate targeting of the SC and dmStriatum. I.e. please confirm that CT cells are projecting to the superior colliculus and the CSt cell are indeed projecting to the striatum.

12) Regarding the silencing result in Figure 5B, it seems as though the CT lesion has affected baseline performance, which seems at odds with the previous Figure 3. Or is the performance at 100% contrast the same? Can the authors test this? The authors suggest that removal of the CT neurons “strongly reduces the impact of VC”. Could the authors put the stats test into the figure here? A paired comparison of all the detection thresholds? Or perhaps copy the curves from 5A into 5B (but dashed)? There is still a reduction in aROC with VC silencing post CT lesion at all datapoints, and this is particularly strong at 100% contrast, where previously there was no effect of silencing. What could the reasons for that be? I think this figure needs to be more clearly explained in the text, it is not as straightforward as it is currently described.

13) The question of what causes this additional reduction leads to the next comment. The 1P silencing was only performed in the context of CT neurons, which raises the question of whether this modulation or sensitivity-enhancement is specific to only this cell type. 1P silencing of the entire VC also shows a similar behavioural effect – is that thought to be predominantly through the action of CT cells? It would be very intriguing to see what the impact of transient silencing does when there are no CSt cells for example.

14) The authors describe the visual stimulus as circular sinusoidal drifting grating displayed on the monitor to capture 30 degrees of the mouse visual field. In the Results section, they write that "a computer monitor was placed to the left visual field of the animal". Yet, it is unclear where exactly the stimulus is in the visual field of the mouse, and whether it is presented only in the left visual field. This needs to be clearly specified as it looks like (although not stated) the viral injections are done unilaterally, and therefore effects on visual performance may vary in the two halves of the visual field (right vs. left). Importantly: where exactly are the cortico-fugal neurons located? The authors state they were mostly found in L5 but do not provide information on exact cortical area (V1 only, other cortices?). Do cortical lesions via viral injections always affect the entire cortical area that corresponds to the retinotopic location of the visual stimulus?

Moreover, the authors use a full field drifting grating stimulus for the experiment depicted in Figure 5 (I assume same is true for Figure 6, although not stated in the Materials and methods). They do not describe the size of the stimulus (in visual degrees) and its exact location in the mouse visual field, and whether it spreads both the left and the right fields. This information can affect the interpretation of the results.

15) Please flesh out the discussion of the roles of the striatum and the superior colliculus, to provide a bit more context, on what is already known of these structures and why they are important to consider in this task. Having these brief discussions will also allow the reader to appreciate the findings a bit more clearly. I.e. it is obviously not novel that the striatum is required for learning (nor is this what the authors claim), so therefore it is not surprising that projections to the striatum are also required for learning. What we learn from this study is that VC provides the most valuable information to striatum, even though it could have come from other sources. Likewise, it is known that SC can subserve a visual task in the absence of V1, though not completely. However, this study suggests that the major influence of VC in this task is mediated through the SC. (though see previous comments on this). In sum, please provide brief summaries with appropriate references to allow readers to position and appreciate these new findings in the existing literature.

---

## [Author Response]

Essential revisions:1) The core of the story is learning; task acquisition and task performance. It is therefore quite surprising that behavioural performance seems so poor – there is a very high false alarm rate even in learned animals (86% probability of lick in the absence of a stimulus, Figure 1 far right example. Note the example in Figure 4 is much better). Could the authors please discuss this issue? While this does not challenge the results in the paper, it does raise some concern over the “expertness” of the animals, i.e. how well they have actually learned. This should be mentioned as a caveat and discussed. What could the possible reasons for the high false alarm rate be? The stimulus duration is quite long at 4 seconds; perhaps with a shorter stimulus (and thus shorter blank period and/or shorter response window) the performance would improve.

The reviewers are correct: The False Alarm rate based on the maximal duration of the response period (4 seconds) is very high even after two weeks of training. Yet, the performance of our animals is far from poor: Reducing the response time window strongly reduces the false alarm rate (Figure 1—figure supplement 2) and, with an optimal response window (i.e. a duration that maximizes Hits and minimizes the False Alarms), our animals have a d-prime value of approximately 3 (i.e. excellent performance) at day 14 of training. As illustrated in Figure 1 and Figure 1—figure supplement 2, within two weeks of training our animals are proficient at the task. For example, the probability of an ideal observer to correctly classify the trial type based on first lick latency is 86±4% (mean ± SEM; N=8 mice).

So, why do we keep such a long response time window? Because the optimal response window varies between training days and between animals, we have developed a scoring metric that is largely independent of the duration of the response window, namely the ROC analysis of first lick latencies. Thus, rather than using a simple Go/NoGo task we specifically adapted the detection task to a “reaction time” based task. In this way, every response, correct or incorrect, has a timestamp (first lick latency), which allows us to characterize the temporal distributions of the responses on stimulus and blank trials using an aROC analysis, a key measure in signal detection theory. While the shortening of the response window during the task reduces the probability of False Alarms (Figure 1—figure supplement 2A), our method offers a highly sensitive approach to reveal sensory guided responses based on the area under the ROC curve. Importantly, this approach obviates the need to make any assumptions about the optimal duration of the response window. In fact, taking advantage of a long response window provides the opportunity to experimentally determine the optimal response window for each individual training session, a fact that we discuss below and now also highlight in the manuscript (Figure 1—figure supplement 2B). Moreover, the measurement of the timing of the behavioral response (i.e. first lick latency), allows us to dissociate two distinct aspects of the behavioral performance: 1) the actual response to the visual stimulus (first lick) and 2) the suppression of licking in the absence of a visual stimulus (inhibitory control). While point 1) is clearly related to the processing of sensory information, point 2) may depend on the internal state of the animal (Figure 1B and Figure 1—figure supplement 2).

Finally, our metric provides an ideal method to characterize the emergence of visually guided behavior. For example, during the initial phases of learning, mice lick randomly at a fast pace in order to obtain water (due to the water restriction) and thus the False Alarm rate is generally as high as the Hit rate (even when considering shorter response periods). However, even at these early stages of learning, one can observe visually evoked responses by comparing the temporal distributions of first lick latencies on stimulus and blank trials. In other words, it is possible to statistically detect visually guided licks by comparing the temporal distributions of the first lick latencies even when the overall amounts of Hits and False Alarms are equal. This is now clearly stated in the manuscript.

2) Further to the previous comment, the authors write “learning was considered complete by 14 days” but it does not appear as though the FA rate has stopped dropping by this time, i.e. the animals are still learning. While the performance in this simple task is not exactly exemplary, it is clear the mice are trying; the concern is how do these results depend on actual task performance, or “expertness”?

Again, as stated above, after 14 days our animals were proficient based on the optimal response window and the aROC metric. To highlight the proficiency of the animals, we also calculated the maximal d-prime for each training day using the optimal response window as described above (d-prime day 14: 3.01±0.32; mean ± SEM, N=8 mice, Figure 1—figure supplement 2B). A d-prime value of around 3 is generally considered as an excellent performance. Furthermore, our ROC analysis shows that an ideal observer would correctly classify 86±4 percent of the trials simply based on first lick latency on day 14 of training. Overall, these data indicated that the animals achieve a very good performance. We considered the learning to have plateaued when aROC was 80% for 4 consecutive days. The slope of the learning curve upon 14 days of training is not significantly different from zero indicating that the learning process in intact animals was complete after 14 days of training. Given that our scoring depends on the temporal distributions of first lick latencies rather than on the overall rate of Hits and False Alarms within a predefined response window, expertness strongly depends on the rapid response upon detection of the visual stimulus rather than on maximizing inhibitory control within a prolonged time window. This is now stated in the Results section and we added panels in Figure 1 and Figure 1—figure supplement 2 to clarify the concerns raised by the reviewers.

3) Wording of main claim: The authors stress throughout the manuscript that they have shown how Str neurons "enhance learning speed" while CT neurons "improve detection sensitivity". Instead what they have shown is that lesioning Str neurons slows down learning and lesioning CT neurons decreases sensitivity. The reversal of interpretation currently used is not justified, since if you claim Str neurons enhance learning speed, it begs the question, enhanced compared to what? We can only make such relative statements from the known rate of learning in an intact animal as a reference, and the correct conclusion is thus an impairment on lesioning. "Impairs the rate of learning", "is required for normal learning rates", etc are alternatives that the authors could use instead.Importantly, the current phrasing leads the reader to assume that the authors have performed a manipulation that actually speeds up learning compared to controls, which is what the Abstract currently implies. I would advise the authors change all phrasing of this kind throughout the entire manuscript, including title and Abstract.

We thank the reviewers for this comment. We have changed our statements throughout the manuscript to match our experimental findings and emphasize that the ablation of selective cortico-fugal neurons leads to specific behavioral impairments during task learning and task performance.

4) Statistics: There are a number problems with the statistics throughout the paper which must be corrected in order to justify the claims.– In Figure 1 it is unclear what statistics have been done to compare the learning rates across conditions. The text only mentions slope, SEM, and n.s. No further information is available in the Materials and methods. This needs to be clarified. What exact test has been done to determine that the slopes in two linear regressions are significantly different? Have the authors calculated the confidence intervals of the slope using bootstrapping? Or have they performed linear regression analysis and measured the significance of an interaction term between session number and lesion condition?

We performed a linear fit of the population average across the mice during the first two weeks of training (i.e. a linear fit over 14 data points) using the Matlab curve fitting toolbox that uses the method of least squares. We determined the standard error and confidence interval of the slope of the linear fit and then used the standard error of the respective slopes to compare the slopes between two experimental groups and determined the statistical significance using the Fisher z-test:z=m1−m2SEm12+SEm22 with m_1_ and m_2_ being the two slopes and SE being the respective standard errors of the linear fits. We now also provide additional information on the learning rate of individual animals as we have fitted the learning curves of individual mice and performed a statistical comparison of the corresponding learning rates across experimental conditions using the Mann Whitney U test (Figure 2D).

– A serious concern is that in Figure 1E, the cortico-striatal lesion condition has been statistically tested against the cortico-tectal, rather than control mice. This does not support the claims made either in text or in the figure, where the control mice plots are overlaid on the same panel.

We thank the reviewers for the comment and have updated the statistical comparisons in the manuscript to represent the proper comparison of CSt-ablated animals against the control group.

– Further, lesion of full visual cortex are compared with striatal lesion and a non-significant difference is reported. Clearly the correct test is to compare VC lesion with the VC controls (which the authors have done when comparing spontaneous lick rates).

We have now added the statistical comparison between VC lesions and VC controls.

– Spontaneous lick rates have been compared in some, but not other cases. This measure is particularly important for the striatal lesion case (see below). Could the authors specify what time periods are included in “spontaneous” and provide comparisons of spontaneous lick rate consistently.

The spontaneous lick rate was assessed during the inter-trial-interval (ITI) period when the monitor was kept gray and licking was not subject to any reinforcement. We added additional data (Figure 2) to highlight the spontaneous lick rates for the experimental groups and updated the most relevant statistical comparisons of the behavioral measures in the main text.

– In a number of cases throughout the manuscript, statistics have been performed on N sessions, which includes multiple sessions from the same animals (e.g. 9 sessions from 6 mice). This is incorrect since multiple sessions from the same mouse are not independent samples. While errors of this kind are unfortunately common in the field at the moment, it is important to avoid it in general, and in particular in this study given that behaviour is highly correlated within individuals. One way to deal with data of this nature is to perform tests with fixed and random effects (e.g. as described in Aarts et al. Nat Neurosci 2014). Another option is to average sessions for each animal.

We thank the reviewers for the comment. We opted to average sessions for each animal and have updated the figures and corresponding analysis to reflect the proper statistical comparison of independent samples.

– Figure 5B: A key claim of this study is that CT lesions reduce sensitivity. However, we only found a comparison of the effect of optogenetic silencing vs. no silencing, but no test between the non-silenced pre vs. post CT lesion. Although the figure legend says “Note reduction in rightward shift of the contrast threshold after CT lesion” this is not actually compared. This seems like a key comparison to support the main claim of the study.

We are sorry for the lack of clarity. With the statement “Note reduction in rightward shift of the contrast threshold after CT-ablation” we wanted to highlight the fact that, while the silencing of visual cortex in intact animals triggers a statistically significant rightward shift of the psychometric function, cortical silencing upon CT-ablation does not lead to a significant increase in the detection threshold. This is the key finding of the experiment, namely that without CT neurons, silencing of visual cortex has little effect on the psychometric function. Originally, we did not perform a statistical test between the psychometric functions in the non-silenced conditions of pre vs. post CT lesion because any rightward shift post CT lesion may be simply due to the three weeks training gap between the two measurements. We now added the corresponding comparison and show that there is indeed a rightward shift. While this shift supports our hypothesis, we also mention the caveats of this comparison.

5) Gross behavioural changes: Measures of gross behavioural changes should be measured with and without lesions in order to rule out the role of these changes in learning rates and sensitivity curves. Importantly, non-stimulus period lick rate should be compared between controls and visual cortex/cortico-striatal lesions, since a reduction in overall lick rate may account for the lower rate of learning.

We agree with the reviewers that changes in learning rates and sensitivity curves could result from differences in gross behavioral changes across groups. However, we did not observe an overall reduction in the spontaneous lick rate upon VC lesion nor upon ablation of CSt neurons or CT neurons (Figure 2D). In addition, all experimental groups performed a similar amount of training trials per day (Figure 2D), indicating that the learning impairments observed upon VC lesion or CSt-ablation are not due to gross behavioral differences.

– The number of trials was limited to 250 per training session: Could the authors clarify if all mice always reached 250 trials in each session, and if not, report the average and range of trials actually performed by the mice in each group. With this information, could the authors rule out that any differences in rates of learning were due to different numbers of trials performed in some days by lesioned mice.

As briefly mentioned above, we limited the amount of training to 250 trials per training session. This limit was based on earlier experiments in which mice were allowed to perform up to 500 trials per training session. We noticed that mice consistently performed the first 250 trials per session across different experimental groups and training days. We have determined the population average of the number of trials per day (Figure 2D) and added the corresponding information in the manuscript.

6) Latency of first licks on blank trials: In the behavioural paradigm used here, latency from blank trials onset is not really a latency, since the mouse experiences a continuous blank screen even though the software might have transitioned from the delay to the “blank” stimulus. This point should be made clear to the reader in the Results and Materials and methods sections, especially when presenting plots like Figure 1B where histograms of lick latencies on blank trials are presented.

Indeed, the latency of the first lick is not an actual latency in relation to an external event as the animal experiences a gray screen throughout the blank trial. We added a schematic of the trial structure (Figure 1—figure supplement 1A, B) and temporal design of the task (Figure 1—figure supplement 1C) to point out how we determined the first lick latency of blank trials.

7) The Introduction is heavily focused on cross species comparisons and evolutionary arguments, and appears more appropriate for a cross species comparative study. In particular, it seems to set up a comparison between mammals, with expanded cortico-fugal pathways, and non-mammals without this expansion. While this is an entirely subjective judgement, this study would benefit from an Introduction more suited to the questions addressed.

We have toned down the cross-species comparison in the Introduction to focus instead on the different cortico-fugal pathways out of VC in the mammalian cortex. Still, we believe that highlighting the fact that the mammalian cortex provides an additional source of sensory information to ancient subcortical structures such as the SC and the striatum, adds an interesting perspective for the readership of our study.

8) Throughout the manuscript, emphases such as the following occur along with essentially each result presented:– "the visual specificity of the licking behavior increased much slower"– "learning progressed much slower and.… was far from complete."– "… strongly reduced the impact"The authors should either substantiate what “much slower”, ”far from complete” etc means, or rephrase as much as possible.

We have removed the emphasis in several of our statements. We have, however, kept some in places where we do not want to hide our surprise or enthusiasm. These remaining statements are now substantiated as requested by the reviewer.

9) It's not immediately clear what the timing of the task is. The schematic in Figure 1A should be extended so as to indicate the timing of the task. The Materials and methods state the stimulus duration is "up to 4 seconds". Why up to? What about the time between trials? What was the actual ITI? Was the no-lick requirement invalidated often for these mice? (do the mice lick constantly or have they learnt to withhold the licking mostly but the 4 second blank is just too long in addition?)

We thank the reviewers for this comment. We added a figure (Figure 1—figure supplement 1) that illustrates in detail trial structure and the timing. The durations of the time periods that constitute a trial varied as a function of training and the exact values are listed in the Materials and methods section and Figure 1—figure supplement 1C. We state that the stimulus duration is “up to 4 seconds” because the duration of the stimulus presentation was contingent on the timing of first lick and thus the stimulus duration varies from one trial to another. We limited the maximum duration of the stimulus presentation or the corresponding blank period to 4 seconds per trial.

The advantages of this response period have been discussed under point 1 and allowed us to sample the temporal distribution of first licks on stimulus and blank trials for the aROC analysis.

Regarding False Alarm rate and the scoring of our task please see extended discussion on point 1 and 2 above.

10) The authors use a metric that is unconventional – aROC of lick latencies. This reaction time metric does make sense and appears convincing from the example in Figure 1. While I like the metric, it is not standard in the field. Therefore it would be nice to more fully compare it to other standard metrics for this type of behaviour – i.e. P(Lick) and d-prime. These comparisons are indirectly available, but It would be better to directly compare a sessions P(Lick) and/or dprime with that session's aROC.

As detailed in point 1 above, we now have an entire figure (Figure 1—figure supplement 2) that compares our analysis with more standard scoring approaches. Furthermore, we would like to highlight that scoring behavior based on response latency is not necessarily unconventional, even though we may be the first to use it to measure performance in a lick/no lick visual detection task in mice. The two main measures to describe psychophysical data are response probabilities and reaction times. A popular method to characterize psychophysical data is the “signal detection theory”, which is based on the Receiver Operating Characteristic (ROC) analysis. In brief, signal detection theory assumes that a behavioral response can be attributed to either a known process (stimulus trial, signal+noise) or be obtained by chance (blank trials, noise only). In most studies, d-prime is calculated by using the z-transform of the Hit rate and the False Alarm rate with the assumptions that: 1) both the signal and noise are normally distributed (with the mean of the noise distribution set to zero), and 2) both distributions have a standard deviation of 1. If so, the d-prime entirely describes the shape of the ROC curve that is created by plotting the Hit rate against the False Alarm Rate across various thresholds.

By taking advantage of a ROC analysis of first lick latencies as a measure to capture the actual temporal distributions of the behavioral response we do not need to make any assumption about the properties of the distribution because the distributions are directly assessed experimentally and thus the ROC analysis provides a nonparametric measure. In fact, we found that the first lick latency distribution on stimulus trials is frequently not normally distributed (e.g. Figure 1B: day1, day 4, day 14; P<0.001, One-sample Kolmogorov-Smirnov test).

11) The bulk of this paper makes use of selective ablation of neurons based on their projection target, via a combination of retroAAV-Cre in the target area and AAV-Caspase3 in the source area. There are no references to existing literature regarding this method: could the authors please add some? It would also strengthen the manuscript if authors could also demonstrate the specificity of this method in at least one of two ways:

Thank you for this comment. We added references of the existing literature on both the retroAAV-Cre and AAV-Caspase3 in the Results.

a) First, how specific is the cell death in the source location? Could the authors provide histology showing that other neurons are indeed spared? What are the consequences of the local death of a good many cells?

We now provide histological data on the effect of cell death of defined cortico-fugal neurons in layer 5 of VC in the main text and quantified the cell density of L5 neurons upon the ablation of CT neurons versus Casp3 only control animals. We observed a reduction of NeuN positive cells in layer 5 which is in line with the cell density of layer 5 CT, suggesting that the cell death is restricted to the targeted neurons. Furthermore, the Casp3 virus used in this study uses a designer pro-Casp3 variant referred to as taCasp3 which triggers cell-autonomous apoptosis, thereby minimizing toxicity to neighboring non-Cre+ cells (Gray et al., 2010; Yang et al., 2013). The relevant references are now included in the manuscript.

Is the function and connectivity of other cells unaffected? (perhaps the literature could provide a hint, these new experiments would be a considerable undertaking. At the least, the potential effects should be discussed).

We have no data relative to whether the selective ablation of one cell group (e.g. CT neurons) affects the connectivity and function of the remaining network. However, given that in several brain regions cortical neurons with distinct projection targets have been shown to form segregated subnetworks, i.e. they mainly synapse among each other (Brown and Hestrin, 2009; Harris and Mrsic-Flogel, 2013; Kim et al., 2018; Lur et al., 2016; Shang et al., 2018; Zhang et al., 2016), we believe that the connectivity and function of the spared neurons is largely unaffected. This is now discussed in the Discussion section.

When the authors describe the “lesion” of CT and CSt, is it a lesion in the same sense as the VC lesions? Perhaps “ablation” is a better word and does more justice to the method.

We thank the reviewers for this comment. We have renamed the experimental groups according to the suggestion: CT-ablation and CSt-ablation throughout the manuscript.

It would be good to see Caspase-only (no Cre) controls, in addition to the Cre-only controls. Does VC remain healthy? Note the concentrations of the virus used are not described in the Materials and methods.

We did not perform Caspase only controls. However, given the specificity of the effect of Caspase injection relative to 1) the ablated cell population as discussed in point 11 (both our data and published data from the Shah lab), and 2) relative to the very different behavioral effects when targeting the CT or the CSt population, we believe that Caspase selectively affects neurons in which it is conditionally expressed. Notably, injection of Caspase to ablate CT before training begins does not impair learning (while Caspase injection to ablate CSt neurons does). This highlights the specificity of the Caspase effect. We have now added the viral concentrations in the Materials and methods section.

b) Second, how accurate are the target site injections? It would be good practice to provide histology confirming accurate targeting of the SC and dmStriatum. I.e. please confirm that CT cells are projecting to the superior colliculus and the CSt cell are indeed projecting to the striatum.

We verify the accuracy of our injection in SC and in the dmSt through the local expression of tdTomato, given that the retroAAV-Cre virus is injected in the Ai14 background. Sections illustrating these injections are now displayed in Figure 1—figure supplement 6. We now also provide a statement in the Materials and methods section. Furthermore, the specificity of our injections is illustrated by the fact that CTB injection in the SC and in the dmSt label non-overlapping (yet intermingled) population of neurons in visual cortex, consistent with the fact that CT and CSt neurons are two separate populations (Figure 1—figure supplement 3). Given that our injections are performed in an Ai14 background, local recombination triggers the expression of tdTomato in SC and dmSt neurons, precluding the visualization of axonal arborization originating from CT and CSt neurons, respectively.

12) Regarding the silencing result in Figure 5B, it seems as though the CT lesion has affected baseline performance, which seems at odds with the previous Figure 3. Or is the performance at 100% contrast the same? Can the authors test this? The authors suggest that removal of the CT neurons “strongly reduces the impact of VC”. Could the authors put the stats test into the figure here? A paired comparison of all the detection thresholds? Or perhaps copy the curves from 5A into 5B (but dashed)? There is still a reduction in aROC with VC silencing post CT lesion at all datapoints, and this is particularly strong at 100% contrast, where previously there was no effect of silencing. What could the reasons for that be? I think this figure needs to be more clearly explained in the text, it is not as straightforward as it is currently described.

As described in point 4 the key finding of the experiment shown in Figure 5B is that without CT neurons, silencing the visual cortex has no significant effect on the psychometric function. We have now performed a statistical test between the psychometric functions in the non-silenced pre vs. post CT lesion and show that there is a significant rightward shift, consistent with the role of the CT projection. However, the performance at 100% contrast is also decreased in the non -silenced condition in post CT-ablation mice as compared to the non-silenced pre lesioned condition, most likely because of the three weeks training gap following the injection of the caspase virus.

We would also like to remind the reviewer that, in contrast to the experiment illustrated in Figure 5B, for those illustrated in Figure 3, the CT-ablation was performed before training onset. For the experiments in Figure 3 we compared the performance at 100% contrast between controls and CT-ablated animals over the entire course of training, i.e. without any training gap. The experiments performed in Figure 5B are now more clearly described in the text (Results).

13) The question of what causes this additional reduction leads to the next comment. The 1P silencing was only performed in the context of CT neurons, which raises the question of whether this modulation or sensitivity-enhancement is specific to only this cell type. 1P silencing of the entire VC also shows a similar behavioural effect – is that thought to be predominantly through the action of CT cells? It would be very intriguing to see what the impact of transient silencing does when there are no CSt cells for example.

This is a good suggestion but beyond the direct scope of the study. By eliminating the rightward shift of the psychometric function in CT-ablated animals upon the cortical silencing of VC, we believe that we have identified the CT pathway as the main source for the rightward shift observed in control animals.

14) The authors describe the visual stimulus as circular sinusoidal drifting grating displayed on the monitor to capture 30 degrees of the mouse visual field. In the Results section, they write that "a computer monitor was placed to the left visual field of the animal". Yet, it is unclear where exactly the stimulus is in the visual field of the mouse, and whether it is presented only in the left visual field. This needs to be clearly specified as it looks like (although not stated) the viral injections are done unilaterally, and therefore effects on visual performance may vary in the two halves of the visual field (right vs. left). Importantly: where exactly are the cortico-fugal neurons located?

There is indeed about a 20 degrees binocular field on each side of the azimuth plane. The center of the stimulus was placed 60 degrees to the left of the azimuth, hence far from the binocular zone. Given that the stimulus radius was 15 degrees, this leaves a margin of 25 degrees between the nasal edge of the stimulus and the temporal edge of the left binocular zone. Thus. we were only stimulating the left, monocular visual field. By stating that the virus was injected in the contralateral cortex relative to the stimulus we were implying unilateral injections. This is now stated more clearly in the main text and the Materials and methods section.

The authors state they were mostly found in L5 but do not provide information on exact cortical area (V1 only, other cortices?). Do cortical lesions via viral injections always affect the entire cortical area that corresponds to the retinotopic location of the visual stimulus?

The viral injections were centered in V1 using 3-4 injections site to forming a triangle (2.3 mm medio-lateral, 0.45 mm anterior from λ; 2.8 mm medio-lateral, 0.45 mm anterior from λ and 2.5 mm medio-lateral, 1 mm anterior from λ) to cover V1 with AAV-Casp3. The ablations covered an area of approximately = 1.4 to 1.9 mm in diameter (Figure 1—figure supplement 5). Thus, the ablations were largely confined to V1 and given V1 visuotopic coordinates certainly covered the stimulated visual area. The surgical cortical lesions affected also visual areas surrounding V1 as the average lesion diameter was about 2.8mm in diameter (Figure 2—figure supplement 1). Thus, the surgical lesion also ablated L5 neurons in these secondary visual cortical areas.

Moreover, the authors use a full field drifting grating stimulus for the experiment depicted in Figure 5 (I assume same is true for Figure 6, although not stated in the Materials and methods). They do not describe the size of the stimulus (in visual degrees) and its exact location in the mouse visual field, and whether it spreads both the left and the right fields. This information can affect the interpretation of the results.

Indeed, a full field drifting grating stimulus was used for both Figures 5 and 6. We updated the Materials and methods section to reflect this accordingly. The full-field stimulus placed in the left (contralateral) visual hemifield, approximately 45 deg from the mouse’s midline at a distance of 14 cm from the left eye, covering approximately 110 degrees of visual space (0° and 110° in azimuth). Thus, while the stimulus was largely covering the left monocular visual field, a portion of the stimulus extended into the binocular zone of the left visual field. However, even animals with bilateral lesion of VC maintained the ability to highly specifically respond to the stimulus (full-field stimulus contrast 100% aROC 0.92±0.026; mean ± SD; N=2 mice), indicating that the behavior does not rely on the visual cortex ipsilateral to the stimulus. These data are now included in the new version of the manuscript (subsection “Visual stimulation”).

15) Please flesh out the discussion of the roles of the striatum and the superior colliculus, to provide a bit more context, on what is already known of these structures and why they are important to consider in this task. Having these brief discussions will also allow the reader to appreciate the findings a bit more clearly. I.e. it is obviously not novel that the striatum is required for learning (nor is this what the authors claim), so therefore it is not surprising that projections to the striatum are also required for learning. What we learn from this study is that VC provides the most valuable information to striatum, even though it could have come from other sources. Likewise, it is known that SC can subserve a visual task in the absence of V1, though not completely. However, this study suggests that the major influence of VC in this task is mediated through the SC. (though see previous comments on this). In sum, please provide brief summaries with appropriate references to allow readers to position and appreciate these new findings in the existing literature.

We thank the reviewer for these comments and suggestions. We expanded our Discussion and also added brief summaries on the function of the SC and the striatum with reference to the existing literature in the Introduction.